# ACTIONS SPEAK LOUDER THAN WORDS: RATE-REWARD TRADE-OFF IN MARKOV DECISION PROCESSES

**Haotian Wu**[*]**, Gongpu Chen**[*][†]**, Deniz Gündüz**
Department of Electrical and Electronic Engineering
Imperial College London,
London SW7 2AZ, U.K.
{haotian.wu17,gongpu.chen,d.gunduz}@imperial.ac.uk

## ABSTRACT

The impact of communication on decision-making systems has been extensively studied under the assumption of dedicated communication channels. We instead consider communicating through actions, where the message is embedded into the actions of an agent which interacts with the environment in a Markov decision process (MDP) framework. We conceptualize the MDP environment as a finite-state channel (FSC), where the actions of the agent serve as the channel input, while the states of the MDP observed by another agent (i.e., receiver) serve as the channel output. Here, we treat the environment as a communication channel over which the agent communicates through its actions, while at the same time, trying to maximize its reward. We first characterize the optimal information theoretic trade-off between the average reward and the rate of reliable communication in the infinite-horizon regime. Then, we propose a novel framework to design a joint control/coding policy, termed *Act2Comm*, which seamlessly embeds messages into actions. From a communication perspective, *Act2Comm* functions as a learning-based channel coding scheme for non-differentiable FSCs under input-output constraints. From a control standpoint, *Act2Comm* learns an MDP policy that incorporates communication capabilities, though at the cost of some control performance. Overall, *Act2Comm* effectively balances the dual objectives of control and communication in this environment. Experimental results validate *Act2Comm*'s capability to enable reliable communication while maintaining a certain level of control performance.

## 1 INTRODUCTION

The role of communication in multi-agent systems has received significant attention as it allows agents with a partial view of the system to better coordinate and cooperate by exchanging messages in parallel to actions taken in the environment (Foerster et al., 2016; Sukhbaatar et al., 2016). However, these systems rely on dedicated channels for communication. On the other hand, explicit communication channels may not always be available, or may be complemented with other *implicit* forms of communication. Such examples are abundant in nature. Bacteria communicate through chemical molecules, known as quorum sensing (Waters & Bassler, 2005), altering their environment and behavior to achieve population-wide coordination. Ants use pheromones to encode the path to food for other ants (von Thienen et al., 2014). Non-verbal communication through gestures, gaze, and even physical appearance, is also known to play an important role in human communication (Trenholm, 2020). In the artificial realm, autonomous robots may also need to rely on implicit communications when explicit communication channels are not available. In medical nano-robots, electromagnetic communication is not feasible due to size and energy limitations, but implicit communication can be achieved through molecular communications (Weiss & Knight, 2001; Wang et al., 2023). Even for more advanced robots, electromagnetic or other types of explicit communication

---

[*]These authors are contributed equally to this work. † Corresponding author.

channels may not be available in harsh or hostile environments; for example, for robots decommissioning nuclear storage facilities, or those operating in deep space, deep ocean, or subterranean environments, e.g., tunnels and caves (Ebadi et al., 2024). Moreover, wireless signals are prone to wiretapping due to their broadcast nature (Poor & Schaefer, 2017), and can be unreliable in adversarial scenarios due to jamming (Pirayesh & Zeng, 2022; Martz et al., 2020), which are other factors limiting explicit communications.

Motivated by these challenges, this paper explores implicit communication in a Markov Decision Process (MDP)—*communication through actions*—which facilitates information transmission from the MDP controller to other agents that can observe the MDP states. Effectively utilizing this internal channel has the potential to reduce the dependence on dedicated communication channels. However, such a communication capability comes with inherent trade-offs. As we will demonstrate, using this channel for communication often leads to a degradation in MDP control performance. This raises a fundamental challenge in balancing control and communication objectives, underscoring the need for a cohesive design that integrates control and communication.

An MDP consists of a controller and an environment (Puterman, 2014). At each time step, the environment is in some state $s$, and the controller selects an action $a$. Upon executing action $a$, the environment stochastically transitions to a new state $s'$ and generates a reward. Controller's objective is to find an optimal policy for selecting an action at each time to maximize its accumulated reward over a given time horizon. Consider that the controller wants to communicate with another agent (i.e., the receiver) that can also observe the environment state.

In communication theory, a finite-state channel (FSC) (Gallager, 1968) is an input-output system with states, where the output depends on both the input and the current state. Messages are encoded into the input sequence, resulting in a corresponding output sequence. The receiver decodes the message from this output sequence. From this perspective, the state transition of an MDP from $s$ to $s'$ upon taking action $a$ can be viewed as an FSC from the controller to the receiver. We refer to this internal channel within the MDP as an *action-state channel*. To communicate through the *action-state channel*, we need an encoder that maps the message to a sequence of actions, and a decoder that translates the resulting state sequence back into the original message. However, the objective of this encoder differs from that of the controller: the encoder aims to maximize the transmission rate and reliability of its message, while the controller seeks to maximize the accumulated reward. These two objectives are generally inconsistent, and a trade-off between the two must be sought.

In this paper, we first investigate the trade-off between the capacity (i.e., the maximum achievable transmission rate) of the *action-state channel* and the MDP reward in the infinite horizon regime. We demonstrate that the capacity of this channel can be expressed in a simple form—as the conditional mutual information between the input and output conditioned on the channel state. We also show that the capacity-reward trade-off can be cast as a convex optimization problem, which can be solved numerically. While the capacity-reward trade-off provides an upper bound on the practically achievable rate under certain reward constraints, solving it does not yield a practical coding scheme. We then propose a practical framework for the integrated control and communication task in the finite block-length regime. The challenge in designing such a framework is twofold: (1) balancing the control and communication performances; and (2) dealing with the non-differentiability of the *action-state channel*. To tackle these issues, we propose *Act2Comm*, a transformer-based coding scheme in which encoder and decoder are trained iteratively.

**Contributions.** The main contributions of this paper are summarized as follows: (1) We introduce a novel paradigm of communicating through actions within an MDP environment, framing it as an integrated control and communication problem. (2) We derive the capacity of the action-state channel, and characterize the capacity-reward trade-off as a convex optimization. (3) We propose *Act2Comm*, a practical transformer-based coding scheme to learn a policy that optimizes communication performance while maintaining a specified level of MDP reward. *Act2Comm* can be of independent interest for designing practical channel coding schemes over other non-differentiable FSC scenarios.

## 2 RELATED WORK

Communication plays a significant role in MDPs, especially in multi-agent reinforcement learning (RL), where agents exchange messages over dedicated or noisy links to achieve a common goal

(Wang et al., 2020; Chen et al., 2024; Tung et al., 2021). This is known as *emergent communications* (Boldt & Mortensen, 2024), but this framework relies on explicit communications over dedicated channels. Implicit communication through actions is considered by Knepper et al. (2017) and Tian et al. (2019). The latter also trains a policy, but it focuses on the multi-agent scenarios, and encourages communication by appropriately changing the reward function. We do not explicitly specify the communicated information, and instead, take a more fundamental approach by characterizing the information theoretic limits of communication and designing a practical coding policy.

Sokota et al. (2022) explored a similar concept of communication via MDPs. In their study, the receiver can observe the entire trajectory, including both the action and state sequences. This enables the controller to encode (compress) messages into the action sequence, and the receiver can subsequently decode the messages from the trajectory. Essentially, this is a source coding problem. However, in most practical scenarios, while the MDP state is a physical signal observable by the receiver, the controller's actions are typically not directly observable by other agents. Therefore, in our work, we assume the receiver can only observe the state sequence. This shifts the problem from source coding to channel coding. Karabag et al. (2019) also examined a similar system, but their focus was on developing policies that restrict the observer's ability to infer transition probabilities.

FSC represents a general class of communication channels, and its study has been a long-standing problem in information and coding theory. Blackwell et al. (1958) studied the capacity of indecomposable FSCs without feedback. Subsequent studies in the non-feedback setting include (Verdu & Han, 1994) and (Goldsmith & Varaiya, 1996). The capacity of FSCs with feedback was examined by Massey (1990) and Permuter et al. (2009). More recently, Shemuel et al. (2022) explored the capacity of FSCs with feedback and state information at the encoder. However, these results express capacity in multi-letter forms, relying on the entire input and output sequences as their lengths approach infinity. Although Sabag et al. (2017) provided a single-letter upper bound for the feedback capacity of unifilar FSCs, exact single-letter expressions for FSC capacity are generally unknown. The action-state channel studied in this paper is a special FSC with state and feedback at the encoder. Utilizing the unique structure of this channel, we derive a single-letter expression for its capacity.

Machine learning has recently advanced traditional channel coding schemes by replacing linear operations with trainable non-linear neural networks, including Turbo autoencoder (Jiang et al., 2019), DeepPolar (Hebbar et al., 2024), KO codes (Makkuva et al., 2021), and other approaches (Jiang et al., 2020; Kim et al., 2018). However, these are designed for Gaussian channels, which are differentiable and allow joint training of the encoder and decoder. Our channel, in contrast, is non-differentiable, presenting new challenges for the design of the encoder and decoder. Channel coding for FSCs is a challenging task with limited results in the literature. Some existing work focuses only on the design of the decoder (Aharoni et al., 2023). However, the main challenge in our problem lies in designing the encoder to balance control and communication performance.

*Notations*: For any sequence $\{x_t : t \geq 1\}$, $\boldsymbol{x_i^k}$ denotes the sub-sequence $\{x_i, x_{i+1}, \ldots, x_k\}$, where $\boldsymbol{x_1^k}$ is written as $\boldsymbol{x^k}$. $|\mathcal{X}|$ denotes the cardinality of the set $\mathcal{X}$. A detailed notation table is provided in Table 1.

## 3 PRELIMINARIES AND SYSTEM MODEL

**Markov Decision Process (MDP).** An MDP can be characterized by a tuple $(\mathcal{S}, \mathcal{X}, \boldsymbol{T}, r, \alpha)$, where $\mathcal{S}$ is the state space, $\mathcal{X}$ is the action space, $\boldsymbol{T}$ is the transition kernel, $r : \mathcal{S} \times \mathcal{X} \to \mathbb{R}$ is the bounded reward function, and $\alpha$ is the initial state distribution. At each time step $t$, taking action $x_t$ in state $s_t$ results in a reward $r(s_t, x_t)$ and a state transition from $s_t$ to $s_{t+1}$, where $s_{t+1}$ is sampled from the distribution $\boldsymbol{T}(\cdot|s_t, x_t)$. We assume both $\mathcal{S}$ and $\mathcal{X}$ are finite sets. A stationary deterministic policy is a mapping $\pi : \mathcal{S} \to \mathcal{X}$ that selects action $x_t$ based on state $s_t$ at each time $t$. The objective is to find an optimal policy that maximizes the long-term average reward, as follows:

$$\textbf{P1:} \quad \max_{\pi} \lim_{N \to \infty} \frac{1}{N} \mathbb{E} \left[ \sum_{t=1}^{N} r(s_t, x_t) | s_1 \sim \alpha \right]. \tag{1}$$

In this paper, we assume that the MDP is unichain; that is, any deterministic policy induces a Markov chain consisting of a single recurrent class plus some transient states. As a result, the optimality of **P1** can be achieved through a stationary deterministic policy. The set of stationary deterministic

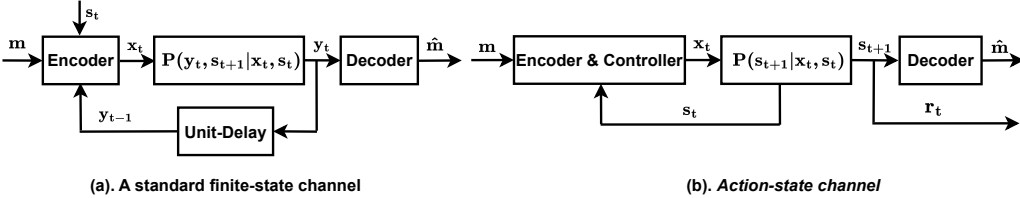

Figure 1: From a standard finite-state channel to an *action-state channel*.

policies is denoted by $\Pi_{SD}$. It is worth noting that the set of admissible policies for an MDP is not restricted to $\Pi_{SD}$. In general, a policy can be history-dependent, determining $x_t$ using all historical states and actions up to time $t$. Let $\Pi_S$ and $\Pi_H$ denote the sets of stationary (possibly randomized) and history-dependent policies, respectively. It is easy to see that $\Pi_{SD} \subset \Pi_S \subset \Pi_H$.

**Finite-State Channel (FSC).** As illustrated in Fig. 1, an FSC can be characterized by a tuple $(\mathcal{X} \times \mathcal{S}, P_{Y,S^+|X,S}, \mathcal{Y} \times \mathcal{S})$, where $\mathcal{X}$ is the input alphabet, $\mathcal{Y}$ is the output alphabet, $\mathcal{S}$ is the channel state alphabet, and $P_{Y,S^+|X,S}$ is the channel law specifying the probability of the channel outputting $Y$ and transitioning to the new state $S^+$, given that the channel input is $X$ in state $S$. We consider a time-invariant channel that exhibits the Markov property, which can be formally expressed as:

$$P(y_t, s_{t+1}|\boldsymbol{x^t}, \boldsymbol{s^t}, \boldsymbol{y^{t-1}}) = P_{Y,S^+|X,S}(y_t, s_{t+1}|x_t, s_t), \; \forall t. \tag{2}$$

To transmit a message $m$, an encoder generates a sequence of inputs $\boldsymbol{x^t}$ as a codeword. When each $x_t$ is input to the channel, the channel transitions to a new state $s_{t+1}$ and produces an output $y_t$. The decoder then collects the output sequence $\boldsymbol{y^t}$ to reconstruct $m$. We suppose that the channel state is available to the encoder but not to the decoder, and that the outputs are fed back to the encoder.

Let $\mathcal{M}$ denote the set of messages, with each message $m$ uniformly sampled from $\mathcal{M}$. The encoder is defined as a sequence of mappings, $\mathcal{E} \triangleq \{\mathcal{E}_1, \ldots, \mathcal{E}_n\}$, where each mapping $\mathcal{E}_t : \mathcal{M} \times \mathcal{S}^t \times \mathcal{X}^{t-1} \times \mathcal{Y}^{t-1} \to \mathcal{X}$ generates the channel input at time $t$. In other words, the channel input at time $t$, $x_t = \mathcal{E}_t(m, \boldsymbol{s^t}, \boldsymbol{x^{t-1}}, \boldsymbol{y^{t-1}})$, is a function of $m$ and all the historical information available at the transmitter up to time $t$. The decoder is defined as the mapping, $\mathcal{D} : \mathcal{Y}^n \to \mathcal{M}$, which reconstructs the message from all $n$ channel outputs, $\hat{m} = \mathcal{D}(\boldsymbol{y^n})$. The pair $(\mathcal{E}, \mathcal{D})$ constitutes a code, where $n$ is called the code length. Suppose the message set is $\mathcal{M} = \{1, 2, 3, \ldots, 2^k\}$, then each message can be represented with $k$ bits. The rate of the code $(\mathcal{E}, \mathcal{D})$ is defined as $R_{\mathcal{E},\mathcal{D}} = k/n$.

The error probability $P_e^{(n)}$ for $(\mathcal{E}, \mathcal{D})$ is defined as $P_e^{(n)} = \Pr(\mathcal{D}(\boldsymbol{y^n}) \neq m | m \text{ is sent})$. A rate $R$ is deemed achievable if there exists a code $(\mathcal{E}, \mathcal{D})$ such that the error probability of the transmission approaches zero as $n \to \infty$. Consequently, the *capacity* of the FSC is defined as the supremum of all achievable rates. In other words, channel capacity reveals the maximum rate available for error-free transmission when the code length approaches infinity. In practice, however, constructing a code with infinite code length is unfeasible. Hence practical channel coding aims to balance the trade-off between $R$ and $P_e^{(n)}$ with a finite code length $n$. For instance, we design codes to maximize the rate while ensuring that the probability of error remains below a certain threshold $\sigma > 0$:

$$\textbf{P2:} \quad \max_{\mathcal{E}, \mathcal{D}} R_{\mathcal{E},\mathcal{D}}, \quad \text{subject to } P_e^{(n)} \leq \sigma. \tag{3}$$

**Integrated Control and Communication.** We investigate a scenario in which the controller of an MDP aims not only to optimize rewards but also to facilitate communication. Assuming that the receiver can observe the state of the MDP, then the environment can be modeled as a specialized FSC, enabling communication between the transmitter (i.e., the controller) and the receiver.

**(a) Action-state channel model:** This integrated FSC is referred to as the *action-state channel*, where the state of the MDP aligns with the state of the FSC. The action and the subsequent state are viewed as the channel input and output, respectively. Upon executing $x_t$ in state $s_t$, the MDP environment returns a reward $r_t$ and transitions to a new state $s_{t+1}$. Here $s_{t+1}$ functions as both the channel output and the new channel state, and $\boldsymbol{s^{t-1}}$ represents not only the historical state sequence, but also the historical feedback signal. The channel law of the *action-state channel* is given by:

$$P(s_{t+1}|\boldsymbol{x^t}, \boldsymbol{s^t}) = P_{S^+|X,S}(s_{t+1}|x_t, s_t) \triangleq \boldsymbol{T}(s_{t+1}|x_t, s_t). \tag{4}$$

This type of channel is also referred to as a POST channel in the literature (Permuter et al., 2014).

**(b) Controller & Encoder:** Within this framework, the MDP controller and the FSC encoder represent the two aspects of the same entity, jointly responsible for selecting an action $x_t \in \mathcal{X}$ at each time step. However, their objectives differ: the controller aims to maximize the reward, while the encoder seeks to maximize the message rate. Unlike the controller, which can focus on stationary deterministic policies, the encoder must account for more complex policy forms. For any message $m$, the encoder described in the previous part can be viewed as a history-dependent policy for the MDP. Therefore, we consider the joint control and coding policy in its most general form. The policy $\mathcal{E}$ is represented as a sequence of mappings $\{\mathcal{E}_i : 0 \leq i \leq n - 1\}$, where $\mathcal{E}_i$ is defined as $\mathcal{E}_i : \mathcal{M} \times \mathcal{S}^i \times \mathcal{X}^{i-1} \to \mathcal{X}$. Each message is transmitted via a sequence of $n$ actions. For example, if the controller begins to transmit a message $m$ at time $t$, then $x_{t+i} = \mathcal{E}_i(m, \boldsymbol{s}_t^{t+i}, \boldsymbol{x}_t^{t+i-1})$ for $0 \leq i \leq n - 1$. We assume the controller always has a new message ready for transmission immediately after completing the transmission of the previous message, and each message is uniformly sampled from $\mathcal{M}$. The long-term average reward of the MDP under policy $\mathcal{E}$ is denoted by $G_{\mathcal{E}}$.

**(c) Decoder:** The receiver observes the state sequence associated with a message and uses it to decode the message. For example, the state sequence associated with the $i$-th message is $\boldsymbol{s}_{in-n+1}^{in}$. The decoder, represented as a mapping : $\mathcal{D} : \mathcal{S}^n \to \mathcal{M}$, decodes the message as: $\hat{m} = \mathcal{D}(\boldsymbol{s}_{in-n+1}^{in})$.

In this paper, we consider non-terminating MDPs over an infinite time horizon and investigate the trade-off between control and communication performance. As discussed previously, if the code length $n \to \infty$, the communication performance can be characterized by the channel capacity (i.e., the maximum achievable rate). Here, we consider a practical setting with finite code length $n$, where the performance of a code $(\mathcal{E}, \mathcal{D})$ is characterized by its rate $R_{\mathcal{E}, \mathcal{D}}$ and the error probability $P_e^{(n)}$. We study the trade-off through the following optimization problem with constants $V$ and $\sigma > 0$,

$$\textbf{P3:} \quad \max_{\mathcal{E}, \mathcal{D}} \ R_{\mathcal{E}, \mathcal{D}} \tag{5}$$

$$s.t. \ G_{\mathcal{E}} \geq V \quad \text{and} \quad P_e^{(n)} \leq \sigma. \tag{6}$$

## 4 The Capacity-Reward Trade-off

In this section, we analyze the trade-off between the capacity of the *action-state channel* and the MDP reward. While the results may not offer direct guidance for practical coding—since the capacity is typically achievable only in the infinite-horizon regime (i.e., when the code length $n \to \infty$)—they delineate the fundamental performance limits of communication via actions in MDPs, thus holding substantial theoretical importance. All proofs of this section are detailed in Appendix B.

In information theory, the capacity of an FSC is usually expressed in terms of conditional mutual information (Shemuel et al., 2024). Let $X, S^+$ and $S$ denote the random variable associated with the input, output (i.e., the next state), and the current state of the *action-state channel*, respectively. Then the conditional mutual information of $X$ and $S^+$ given $S$ is defined as (Cover, 1999):

$$I(X; S^+|S) = \mathbb{E}_{p(x, s^+, s)} \left[ \log \frac{p(s, s^+|s)}{p(x|s)p(s^+|s)} \right]. \tag{7}$$

Let $\pi(\cdot|s)$ denote an input distribution of the channel given that the channel state is $s \in \mathcal{S}$. For a given channel, the joint distribution $p(x, s^+, s)$ is determined by the conditional input distribution $\pi$. From the MDP perspective, $\pi(x|s)$ represents the probability of selecting action $x$ in state $s$; thus, $\pi$ can be viewed as a stationary randomized policy for the MDP. Let $\rho_\pi$ denote the equilibrium state distribution of the MDP under policy $\pi$. We have the following result:

**Theorem 1** *The capacity of the action-state channel without reward constraint is given by*

$$C = \max_{\{\pi(x|s): x \in \mathcal{X}, s \in \mathcal{S}\}} I(X; S^+|S)$$

*where $X, S,$ and $S^+$ follow a joint distribution given by*

$$p(x, s^+, s) = \rho_\pi(s)\pi(x|s)\boldsymbol{T}(s^+|s, x), \ x \in \mathcal{X}, s, s^+ \in \mathcal{S}.$$

As previously discussed, a general encoder for an FSC generates a channel input based on all the historical state and feedback information. However, Theorem 1 reveals a surprising fact: the capacity of the *action-state channel* can be achieved by encoding messages using a stationary randomized policy for the MDP, without relying on historical information.

Theorem 1 presents the capacity of the *action-state channel* without considering the MDP reward. If we want to maintain a certain level of long-term average reward for the MDP, the capacity may generally decrease. Next, we characterize the trade-off between channel capacity and MDP reward.

Given a stationary policy $\pi$ for the MDP, define $w_\pi(s, x) = \rho_\pi(s)\pi(x|s)$. Here, $w_\pi(s, x)$ represents the long-term proportion of time that the MDP is in state $s$ and takes action $x$. In the literature, $w_\pi$ is referred to as the occupation measure of policy $\pi$ (Altman, 2021). Let $\mathcal{W}$ denote the set of all occupation measures, then $\mathcal{W}$ is the set of $w \in \mathbb{R}^{|\mathcal{S}| \times |\mathcal{X}|}$ satisfying the following equations:

$$\sum_{x \in \mathcal{X}} w(s, x) - \sum_{s' \in \mathcal{S}} \sum_{x' \in \mathcal{X}} w(s', x')\boldsymbol{T}(s|s', x') = 0, \ \forall s \in \mathcal{S}, \tag{8}$$

$$\sum_{s \in \mathcal{S}} \sum_{x \in \mathcal{X}} w(s, x) = 1, \ w(s, x) \geq 0, \ \forall s \in \mathcal{S}, x \in \mathcal{X}. \tag{9}$$

Clearly, $\mathcal{W}$ is a polytope. It is well-known that there is a one-to-one mapping between $\mathcal{W}$ and $\Pi_S$. In particular, $\pi(x|s) = w_\pi(s, x)/\sum_{x'} w_\pi(s, x')$ for any $s \in \mathcal{S}, x \in \mathcal{X}$. Using this relationship, the problem of computing the capacity with reward constraint $V$ (i.e., ensuring the long-term average reward is not less than $V$) reduces to a convex optimization, as stated in the following theorem:

**Theorem 2** *The capacity of the action-state channel with reward constraint $V$ is the optimal value of the following convex optimization problem:*

$$\max_{w \in \mathcal{W}} I(w, \boldsymbol{T})$$

$$s.t. \sum_{s \in \mathcal{S}} \sum_{x \in \mathcal{X}} w(s, x)r(s, x) \geq V$$

*where $I(w, \boldsymbol{T})$ is a concave function of $w \in \mathcal{W}$ defined as*

$$I(w, \boldsymbol{T}) \triangleq \sum_{s \in \mathcal{S}} \sum_{x \in \mathcal{X}} w(s, x) \sum_{s' \in \mathcal{S}} \boldsymbol{T}(s'|s, x) \log \frac{\boldsymbol{T}(s'|s, x) \sum_{x''} w(s, x'')}{\sum_{x'} \boldsymbol{T}(s'|s, x')w(s, x')}.$$

Denote by $C(V)$ the capacity of the *action-state channel* with reward constraint $V$.

**Lemma 1** $C(V)$ *is a concave function.*

Since the capacity is an upper bound for the rate of any practical coding scheme, Lemma 1 implies that the achievable region of rate-reward pairs forms a convex set.

The convex optimization problem in Theorem 2 can be efficiently solved using the gradient ascent algorithm if the gradient of the objective function has a closed-form expression (Bertsekas, 2016). Next, we derive the gradient of $I(w, \boldsymbol{T})$ with respect to $w$. Define

$$l(w, w_n, \boldsymbol{T}) \triangleq \sum_{s \in \mathcal{S}} \sum_{x \in \mathcal{X}} w(s, x) \sum_{s' \in \mathcal{S}} \boldsymbol{T}(s'|s, x) \log \frac{\boldsymbol{T}(s'|s, x) \sum_{x''} w_n(s, x'')}{\sum_{x'} \boldsymbol{T}(s'|s, x')w_n(s, x')}, \ w, w_n \in \mathcal{W}.$$

**Lemma 2** *For any $w_n \in \mathcal{W}$, $l(w, w_n, \boldsymbol{T})$ is a tangent line of $I(w, \boldsymbol{T})$ at point $w_n$. That is,*

*(i)* $l(w_n, w_n, \boldsymbol{T}) = I(w_n, \boldsymbol{T})$.

*(ii)* $l(w, w_n, \boldsymbol{T}) \geq I(w, \boldsymbol{T})$ *for all $w$.*

It follows immediately from Lemma 2 that

$$\nabla I_{w_n}(s, x) \triangleq \left.\frac{\partial I(w, \boldsymbol{T})}{\partial w(s, x)}\right|_{w=w_n} = \sum_{s' \in \mathcal{S}} \boldsymbol{T}(s'|s, x) \log \frac{\boldsymbol{T}(s'|s, x) \sum_{x''} w_n(s, x'')}{\sum_{x'} \boldsymbol{T}(s'|s, x')w_n(s, x')}, \tag{10}$$

for any $w_n \in \mathcal{W}$, $s \in \mathcal{S}$, and $x \in \mathcal{X}$. The gradient $\partial I/\partial w = [\nabla I_w(s, x)]_{s,x}$ then can be used in the gradient ascent method to solve the optimization problem in Theorem 2.

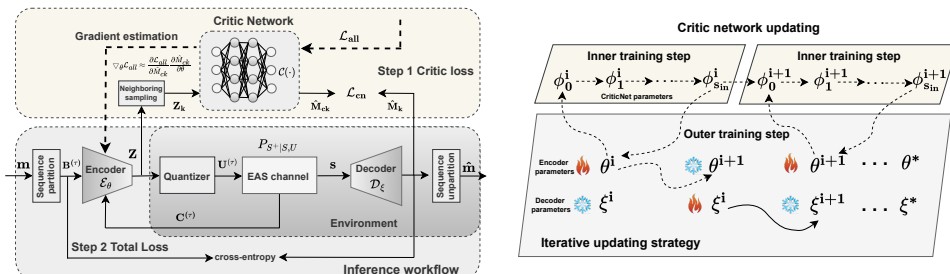

Figure 2: (Left) Workflow diagram of the *Act2Comm* scheme, with the dashed line indicating the gradient flow. (Right) Illustration of the iterative training strategy, incorporating a critic network.

## 5 ACT2COMM: A PRACTICAL CODING SCHEME

This section presents *Act2Comm*, a learning-based practical coding scheme that balances both control and communication objectives. This framework assumes a pre-determined control policy $\pi$ that satisfies the reward constraint $G_\pi \geq V$, referred to as the target policy. Such a target policy can be easily derived using traditional MDP or RL algorithms. Alternatively, solving the problem in Theorem 2 yields a policy $\pi$ that precisely satisfies $G_\pi = V$. *Act2Comm* aims to learn a coding policy that: (1) closely mimics the stochastic behavior of the target policy; (2) minimizes the probability of decoding errors for a given coding rate and a finite code length. That is, *Act2Comm* takes a policy achieving the desired reward, and embeds messages into it with the desired reliability. For ease of reference, we denote the element of matrix $\boldsymbol{X}$ at the $i$-th row and $j$-th column as $X[i,j]$. The sets of states and actions are indexed as $\mathcal{S} = \{0, 1, \ldots, |\mathcal{S}| - 1\}$ and $\mathcal{X} = \{0, 1, \ldots, |\mathcal{X}| - 1\}$.

**Channel transform.** Let $\mathcal{U} \triangleq \mathcal{X}^{|\mathcal{S}|}$. Each $u \in \mathcal{U}$ is referred to as a decision rule as the $i$-th element of $u$ can be viewed as an action prescribed for state $i$. A deterministic control policy can be defined as a sequence $\{u_t : t \geq 1\}$, where $u_t \in \mathcal{U}$ is the decision rule at time $t$. To facilitate block coding, we first convert the *action-state channel* into an extended action-state (EAS) channel (Fig. 6 in Appendix) using Shannon's method (Shannon, 1958). We conceptually separate the encoder and controller, considering the controller as part of the EAS channel. Channel state is assumed to be available at the controller, but not the encoder. At each time $t$, the encoder selects a decision rule $u_t$ from $\mathcal{U}$. Then the controller uses $u_t$ and the state $s_t$ to determine an action $x_t = u_t(s_t)$. Consequently, the EAS channel has input alphabet $\mathcal{U}$, output alphabet $\mathcal{S}$, and channel law:

$$P_{S^+|S,U}(s_{t+1}|s_t, u_t) = P_{S^+|S,X}(s_{t+1}|s_t, u_t(s_t)) \triangleq \boldsymbol{T}(s_{t+1}|s_t, u_t(s_t)). \tag{11}$$

The EAS channel and the *action-state channel* are equivalent. We will focus on the former to develop our coding scheme. This approach allows us to map a data block to a sequence of decision rules without knowing the future states. Actions for subsequent time steps can then be determined using these decision rules when the states are revealed. In the rest of this section, we detail the *Act2Comm* framework, which consists of five components.

**1) Overall workflow.** As depicted in Fig. 2, given a $k$-bit message $\boldsymbol{m} \in \{0, 1\}^k$, the *Act2Comm* first encodes $\boldsymbol{m}$ into a belief map $\boldsymbol{Z} \in \mathbb{R}^{|\mathcal{S}| \times \frac{k}{R}}$. This $\boldsymbol{Z}$ is subsequently mapped into a codeword $\boldsymbol{U} \in \mathcal{X}^{|\mathcal{S}| \times \frac{k}{R}}$ by a quantizer. At each time step $t$, the controller selects an action $x_t = \boldsymbol{U}[s_t, t]$ for state $s_t$, and the channel transitions into a new state according to the channel law $P_{S^+|S,X}$. After $\frac{k}{R}$ time steps, the receiver decodes the message based on the accumulated observations $\boldsymbol{s} \in \mathcal{S}^{\frac{k}{R}}$.

**2) Block-attention feedback coding.** One of the principal innovations of *Act2Comm* is the *block-attention* coding mechanism, which reduces the coding complexity and enhances the performance.

**(a) Message block:** Formally, we partition message $\boldsymbol{m}$ into $l$ blocks as $\boldsymbol{m} = [\boldsymbol{b_1}; \boldsymbol{b_2}; \ldots; \boldsymbol{b_l}]$, where each block $\boldsymbol{b_i} \in \{0, 1\}^\mu$ contains $\mu = k/l$ bits. This allows us to encode $\boldsymbol{m}$ with $l$ coding rounds, with each round consisting of $\mu/R$ time steps. For each coding round $\tau$ ($1 \leq \tau \leq l$), the input message block is defined as $\boldsymbol{B}^{(\tau)} \triangleq [2\boldsymbol{b_1} - 1; \ldots; 2\boldsymbol{b_\tau} - 1] \in \mathbb{R}^{\tau \times \mu}$.

**(b) Feedback block:** Although Theorem 1 shows that the capacity-achieving code with infinite blocklength can be history-independent, feedback has demonstrated benefits in simplifying the coding process for better performance in the practical finite blocklength regime (Kostina et al., 2017; Kim et al., 2020). Hence, for each time step $t$ within the $\tau$-th coding round, we introduce the feedback vector as $\boldsymbol{c_t^{(\tau)}} \triangleq [s_t^{(\tau)}, x_t^{(\tau)}, s_{t+1}^{(\tau)}] \in \mathbb{R}^{1\times 3}$, which encapsulates the current state, the selected action, and the subsequent state. The feedback matrix for the $\tau$-th round can then be given by $\boldsymbol{C_\tau} = [\boldsymbol{c_1^{(\tau)}}; \ldots; \boldsymbol{c_{\frac{\mu}{R}}^{(\tau)}}] \in \mathbb{R}^{\frac{\mu}{R}\times 3}$. Consequently, for each coding round $\tau$, we concatenate prior feedback matrices to construct a feedback block: $\boldsymbol{C^{(\tau)}} \triangleq [\boldsymbol{C_1}; \ldots; \boldsymbol{C_\tau}] \in \mathbb{R}^{\tau \times \frac{\mu}{R}\times 3}$.

### 3) Transceiver design.

**(a) Encoder:** At each coding round $\tau$, a transformer-based encoder is utilized to generate a belief matrix $\boldsymbol{Z^{(\tau)}} \in \mathbb{R}^{\tau \times \frac{\mu|\mathcal{S}|}{R}}$ using $\boldsymbol{B^{(\tau)}}$ and $\boldsymbol{C^{(\tau)}}$. The detailed architecture is provided in Fig. 8b, with each component illustrated in Appendix C. The $\tau$-th row vector of $\boldsymbol{Z^{(\tau)}}$, denoted as $\boldsymbol{z^{(\tau)}} \triangleq \boldsymbol{Z^{(\tau)}}[\tau,:] \in \mathbb{R}^{\frac{\mu|\mathcal{S}|}{R}}$, represents the belief vector derived from the $\tau$-th coding round. After completing all $l$ coding rounds, the selected belief vectors are combined to form the final belief map $\boldsymbol{Z} = [\boldsymbol{z^{(1)}}; \ldots; \boldsymbol{z^{(l)}}] \in \mathbb{R}^{\frac{k}{\mu}\times \frac{\mu|\mathcal{S}|}{R}}$, which is subsequently reshaped into $\boldsymbol{Z} \in \mathbb{R}^{|\mathcal{S}|\times \frac{k}{R}}$. Each element of this reshaped belief map $Z[s_t, t]$ indicates the action belief at time step $t$ given state $s_t$.

**(b) Quantizer:** *Act2Comm* employs a quantizer to generate the codeword $\boldsymbol{U} \in \mathcal{X}^{|\mathcal{S}|\times \frac{k}{R}}$ as:

$$\boldsymbol{U} = \mathcal{Q}(|\mathcal{X}| \cdot \text{Sigmoid}(\boldsymbol{Z})), \tag{12}$$

where $\mathcal{Q} : \mathbb{R}^{|\mathcal{S}|\times \frac{k}{R}} \rightarrow \mathcal{X}^{|\mathcal{S}|\times \frac{k}{R}}$ is the quantization operation that maps the coding result to the nearest action index in the action space $\mathcal{X}$, and each element of resultant codeword, $x_t = \boldsymbol{U}[s_t, t]$, represents the selected action for state $s_t$ at time step $t$.

**(c) Decoder:** Given the state observations $\boldsymbol{s} \triangleq [s_1; \ldots; s_{\frac{k}{R}}] \in \mathcal{S}^{\frac{k}{R}}$, a transformer-based decoder is utilized to output logits $\hat{\boldsymbol{M}} \in \mathbb{R}^{\frac{k}{\mu}\times 2^\mu}$ for all blocks. After applying the softmax function, each block is predicted and subsequently transformed into the reconstructed bitstream $\hat{\boldsymbol{m}}$.

**4) Joint optimization of control and communication.** To model the trade-off between control and communication, we utilize a weighted loss function to train the encoder: $\mathcal{L}_{all} = \mathcal{L}_{com} + \lambda \mathcal{L}_{cont}$. The communication loss $\mathcal{L}_{com}$ is defined as the cross-entropy between the predictions from a critic network and their corresponding ground-truth, which quantifies the message decoding accuracy. To ensure control performance, we aim to make the coding policy behave closely to the target policy. Therefore, $\mathcal{L}_{cont}$ measures the "distance" between the coding policy and the target policy.

Let $\pi$ denote the target policy, with $\pi(x|s)$ representing the probability of taking action $x$ in state $s$. Let $f_U(x|s)$ denote the frequency of selecting $x$ in state $s$ across all decision rules in $\boldsymbol{U}$. We then use the mean square error (MSE) between $\pi$ and $f_U$ to measure the control loss for its stability in experiments. However, $f_U$ is non-differentiable during the backpropagation as it is discrete. To address this issue, we estimate $f_U(x|s)$ using $\boldsymbol{Z}$ in equation 12. Let $\boldsymbol{e}$ denote the all-one row vector, and define $\Gamma_Z(\boldsymbol{T}, s, x) \triangleq \text{Sigmoid}(\gamma(|\mathcal{X}|\text{Sigmoid}(\boldsymbol{Z}[s,:]) - x\boldsymbol{e}))$. When $\gamma > 0$ is sufficiently large, $\Gamma_Z(\boldsymbol{T}, s, x)$ is a $(kR)$-dim vector with elements close to either 0 or 1. Additionally, $\Gamma_Z(\boldsymbol{T}, s, x)\boldsymbol{e}^\top$ approximates the number of elements in $\boldsymbol{U}[s,:]$ that are not less than $x$. We refer to $\gamma$ as the temperature parameter and estimate $f_U(x|s)$ for $x > 0$ as follows:

$$f_U(x|s) \approx \hat{f}_U(x|s) \triangleq \frac{1}{kR}\left[\Gamma_Z(\boldsymbol{T}, s, x-1)\boldsymbol{e}^\top - \Gamma_Z(\boldsymbol{T}, s, x)\boldsymbol{e}^\top\right]. \tag{13}$$

For $x = 0$, we have $f_U(0|s) \approx \hat{f}_U(0|s) \triangleq 1 - \Gamma_Z(\boldsymbol{T}, s, 0)\boldsymbol{e}^\top / kR$. As a result, we define the control loss as $\mathcal{L}_{cont} = MSE(\pi, \hat{f})$.

**5) Iterative training strategy.** Given the non-differentiable nature of the EAS channel and quantizer, jointly updating the encoder and decoder is infeasible. To address this, we introduce a critic network and employ an iterative updating strategy to train *Act2Comm* effectively, with the corresponding algorithm and architectures detailed in Appendix C.2.

**(a) Critic network.** As shown in Fig. 2, a critic network is introduced to estimate the gradient during gradient backpropagation for the encoder optimization, which views the EAS channel and decoder

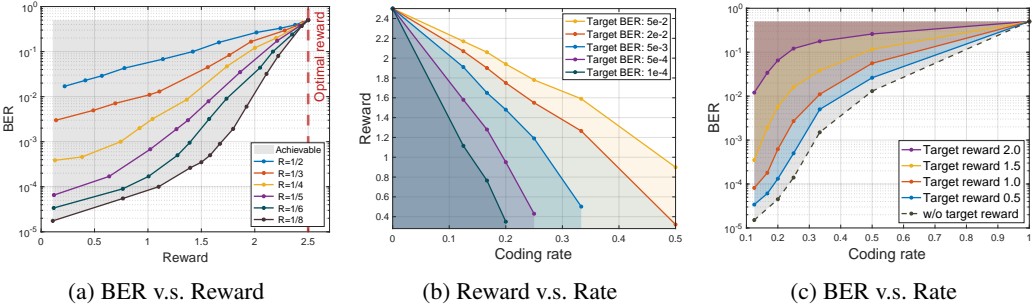

| (a) BER v.s. Reward | (b) Reward v.s. Rate | (c) BER v.s. Rate |
|---|---|---|

Figure 3: Control-communication trade-off of *Act2Comm* in "Lucky Wheel".

as an unknown environment. Before each update of the encoder, a critic network is trained over $s_{in}$ inner steps to predict the logits for the neighbor belief maps of a given $\mathbf{Z}$. For each inner step $k$, the network is trained to predict the corresponding logits $\hat{\mathbf{M}}_{\mathbf{k}}$ as $\hat{\mathbf{M}}_{\mathbf{ck}}$ based on the neighbor belief maps sampled from $\mathbf{Z_k} = \mathbf{Z} + \mathbf{W_k}$, where $\mathbf{W_k} \in \mathbb{R}^{|\mathcal{S}| \times \frac{k}{R}} \sim \mathcal{N}(0, \sigma_w^2)$ is the Gaussian noise term for neighboring sampling during the $k$-th inner step. The MSE loss, denoted as $\mathcal{L}_{cn}$, is utilized between $\hat{\mathbf{M}}_{\mathbf{ck}}$ and $\hat{\mathbf{M}}_{\mathbf{k}}$ to train the critic network. With this design, we aim to obtain a precise critic network to estimate gradients from neighbors of $\mathbf{Z}$ in a given environment, thereby helping update the encoder. Note that this extra training cost is incurred only during the offline training process, this critic network will be removed during the inference phase, as detailed in Appendices C.3-C.4..

**(b) Iterative updating strategy.** As outlined in Fig. 2, at each update step $i$, we first train a critic network $\phi_{s_{in}}^i$ using $s_{in}$ inner steps to learn to estimate the gradient around the samples. Next, the encoder is updated to $\theta^{i+1}$ using the frozen decoder parameters $\xi^i$ and the learned gradient estimation. Subsequently, the decoder is directly optimized with the loss function to obtain new parameters $\xi^{i+1}$, while keeping the encoder frozen at $\theta^{i+1}$. This process iteratively alternates between encoder and decoder updates, freezing one while optimizing the other at each step.

# 6 EXPERIMENTAL RESULTS

We evaluate *Act2Comm* across three distinct MDP environments, as detailed in Appendix D, with communication performance measured by the bit error rate (BER). Due to page limitations, the results of the third environment, "Erratic robot", are provided in the Appendix D.4.

**Experiment 1: Lucky Wheel.** In this game, the agent keeps spinning a wheel to accumulate rewards by choosing either clockwise or counterclockwise direction. It is modeled as an MDP with 3 states and 2 actions, the details of the environment and experimental setting are provided in Appendix D.

We examine the trade-off among the three performance metrics in Fig. 3. We set the optimal reward-maximizing policy as the target policy, and consider different code rates. By adjusting $\lambda$, we can control how closely the coding policy approximates the target policy. The shaded regions in figures (a)-(c) represent achievable regions of *Act2Comm* with various $\lambda$. When $\lambda$ is large, regardless of the coding rate, *Act2Comm* learns a policy that mirrors the target policy. In this case, all messages are mapped to the same sequence of decision rules since the target policy is stationary and deterministic. As a result, the BER is 0.5, indicating zero communication capability.

Next, we consider different target BERs, resulting in a trade-off between the code rate and reward. As shown in Fig. 3b, achieving a pre-determined BER with a higher coding rate results in a reduced reward. When targeting a lower BER, the reward decreases rapidly with the coding rate. Fig. 3c illustrates *Act2Comm*'s ability to balance BER and coding rate when ensuring a specified reward. Our results reveal that reducing the reward constraint leads to more reliable communication at the same rate. In summary, these findings demonstrate that *Act2Comm* can communicate messages through its actions at acceptable reliability while satisfying specific reward criteria.

**Experiment 2: Catch the Ball.** We next evaluate *Act2Comm* in "Catch the Ball", which is an MDP with 27 states and 3 actions. This MDP includes a parameter $p$ that influences its transition matrix

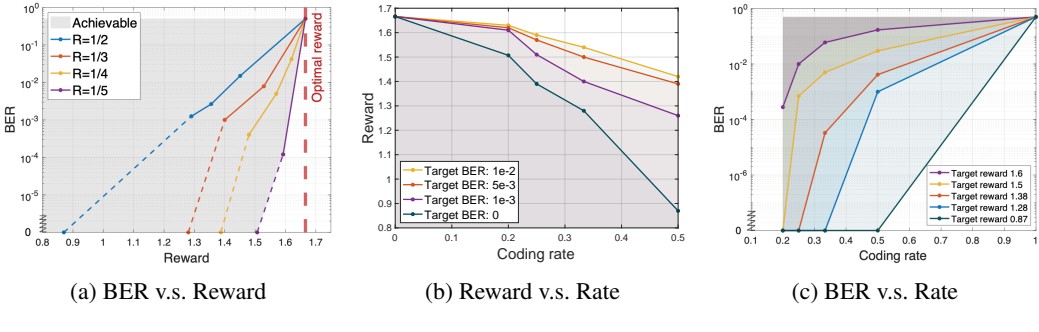

(a) BER v.s. Reward      (b) Reward v.s. Rate      (c) BER v.s. Rate

Figure 4: Control-communication trade-off of *Act2Comm* in "Catch the Ball" with $p = 0$.

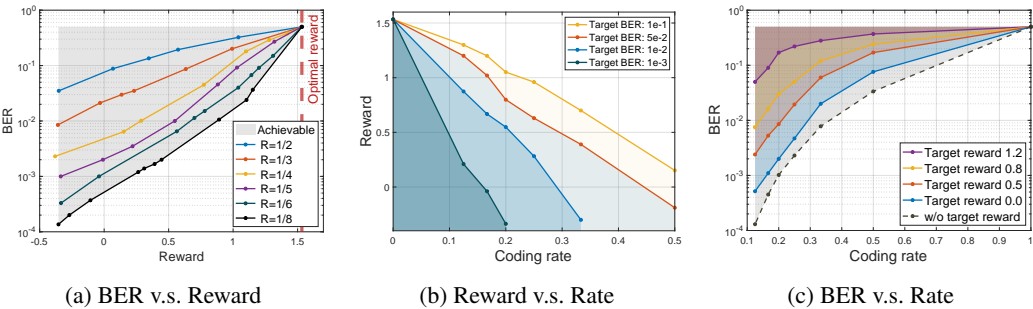

(a) BER v.s. Reward      (b) Reward v.s. Rate      (c) BER v.s. Rate

Figure 5: Control-communication trade-off of *Act2Comm* in "Catch the Ball" with $p = 0.2$.

(see Appendix D for details). When $p = 0$, the action-state channel is perfect as each action can be reliably inferred from the resulting state transition, but it becomes noisy for $p > 0$.

We first consider $p = 0$, where no coding is needed since there is no noise. The challenge is to maintain a certain reward while communicating. *Act2Comm* performs excellently in this environment. As shown in Fig. 4a, with a minor reduction in reward from $1.66$ to $1.5$, *Act2Comm* communicates at a rate of $0.2$ with no error. If we relax BER to $10^{-4}$, same rate can be achieved with a reward of $1.6$. The trade-off between reward and rate is shown in Fig. 4b. This experiment highlights our model's capacity to enable efficient communication capabilities with minimal impact on performance. We also applied *Act2Comm* to this game with $p = 0.2$, in which the action-state channel is noisy and the coding process becomes more complex. As detailed in Fig. 5, we can observe a reduction in the coding rates for the same level of reliability due to the stochasticity in the environment.

## 7    CONCLUSION

We introduced a novel framework of *communication through actions*, a form of implicit communication from the controller of an MDP to a receiver that can observe the states. By treating the MDP environment as a communication channel, messages can be encoded into the action sequence and decoded from the state sequence. Aiming to optimize communication performance while ensuring a certain MDP reward, we formulated an integrated control and communication problem. We derived the capacity of the action-state channel and demonstrated that the trade-off between channel capacity and reward can be characterized as a convex optimization problem. We then proposed *Act2Comm*, a transformer-based framework for designing joint control and communication policies. Through experiments, we demonstrated *Act2Comm*'s capability to communicate reliably through actions while maintaining a certain level of MDP reward.

The proposed *Act2Comm* framework can be used as a plug-in component in various MDP and RL applications, enabling information transmission by learning a joint control and coding policy that closely mimics the target policy. More importantly, our study demonstrates the potential of communication through actions in multi-agent systems. While this form of implicit communication leads to some loss in control performance, it may potentially improve the overall control performance by enhancing coordination when applied to multi-agent systems where explicit communication channels are not available. This presents an interesting and challenging direction for future research.

ACKNOWLEDGMENTS

We acknowledge funding from the UKRI for the projects AI-R (ERC Consolidator Grant, EP/X030806/1) and INFORMED-AI (EP/Y028732/1), as well as the SNS JU project 6G-GOALS under the EU's Horizon program (Grant Agreement No. 101139232).

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

## APPENDIX

## A  NOTATION AND DEFINITIONS

To bridge the RL and FSC areas, we list and explain the notations used in this paper. Note that lowercase and uppercase bold letters represent vectors and matrices, respectively.

Table 1: Notation table

| | |
|---|---|
| **In general** | |
| $\boldsymbol{x_i^k}, \boldsymbol{x^k}$ | Sequence $\{x_i, x_{i+1}, \ldots, x_k\}$, sequence $\{x_1, \ldots, x_k\}$ |
| $\lvert \mathcal{X} \rvert$ | Cardinality of set $\mathcal{X}$ |
| **MDP: $(\mathcal{S}, \mathcal{X}, \boldsymbol{T}, r, \alpha)$** | |
| $s_t, x_t$ | MDP state and action at $t$ |
| $r, \alpha$ | Reward function and initial state distribution |
| $\mathcal{S}, \mathcal{X}$ | State and action space for MDP |
| $\boldsymbol{T}, \pi$ | Transition kernel and control policy |
| **FSC: $(\mathcal{X} \times \mathcal{S}, P_{Y,S^+\lvert X,S}, \mathcal{Y} \times \mathcal{S})$** | |
| $x_t, y_t, s_t$ | Channel input, output, and state at $t$ |
| $\mathcal{X}, \mathcal{Y}, \mathcal{S}$ | Channel input, output, and state alphabets |
| $S^+, S$ | Future and current state (random variables) |
| $\mathcal{E}, \mathcal{D}$ | Encoder and decoder |
| $k, n$ | Message bit length and the code length |
| $R_{\mathcal{E},\mathcal{D}} = k/n$ | Rate of code $(\mathcal{E}, \mathcal{D})$ |
| **Action-state channel: $P(s_{t+1}\lvert \boldsymbol{x^t}, \boldsymbol{s^t})$** | |
| $s_t, x_t$ | State and action at $t$ |
| $\mathcal{S}, \mathcal{X}$ | State alphabet, action alphabet |
| $S^+, S$ | Future and current state (random variables) |
| $\mathcal{E}, \mathcal{D}, \boldsymbol{T}, G_{\mathcal{E}}$ | Encoder, decoder, transition kernel, average reward |
| $k, n, R$ | Message bit length, code length, and rate |
| $\boldsymbol{s}_{in-n+1}^{in}$ | The state sequence associated with the $i$-th message |
| **EAS channel for *Act2Comm*: $P(s_{t+1}\lvert \boldsymbol{u^t}, \boldsymbol{s^t})$** | |
| $s_t, x_t, u_t; \pi$ | State, action and decision rule at $t$; Target policy |
| $S^+, S$ | Future and current state (random variables) |
| $\mathcal{U}, \mathcal{X}, \mathcal{S}$ | Alphabets of decision rule, actions, and states |
| $\boldsymbol{T}, G_{\mathcal{E}}$ | transition kernel and average reward |
| $\mathcal{E}(\cdot), \mathcal{D}(\cdot), \mathcal{C}(\cdot)$ | Encoder, decoder, and critic network. |
| $\boldsymbol{\theta}, \boldsymbol{\xi}, \boldsymbol{\phi}$ | Parameters of encoder, decoder, critic network |
| $\boldsymbol{Z}, \boldsymbol{U}$ | Encoded belief map and codeword |
| $\boldsymbol{B}^{(\tau)}, \boldsymbol{C}^{(\tau)}$ | Message and feedback block at coding round $\tau$ |
| $\boldsymbol{X}^{(\tau)}$ | Control policy at coding round $\tau$ |
| $\hat{\boldsymbol{M}}$ | Decoded logits |
| $\mathcal{Q}(\cdot)$ | Quantizer |
| $\mathcal{L}_{all}$ | Weighted loss function to update the encoder |
| $\mathcal{L}_{com}, \mathcal{L}_{cont}$ | Communication loss term and Control loss term |
| $\mathcal{L}_{cn}$ | MSE loss for critic network |
| $f_U(x\lvert s)$ | Frequency of selecting action $x$ in state $s$ across $\boldsymbol{U}$ |
| $\gamma$ | Estimation temperature for control policy |
| $\hat{f}_U(x\lvert s)$ | Estimation of $f_U(x\lvert s)$ |
| $\boldsymbol{Z_k}, \boldsymbol{W_k}$ | Sampled neighbors of $\boldsymbol{Z}$ and its Gaussian term |
| $\hat{\boldsymbol{M}_k}$ | The corresponding logits from the frozen decoder |
| $\hat{\boldsymbol{M}_{ck}}$ | Predicted logits from the critic network (the $k$-th step) |

Figure 6: The *action-state channel* and the equivalent *extended action-state channel*.

## B  TECHNICAL PROOFS

This section presents the proofs of Section 4.

### B.1  PROOF OF THEOREM 1

The proof of Theorem 1 relies on converting the action-state channel to an equivalent channel. This equivalence is also stated in Section 5, as it is crucial for the design of *Act2Comm*. To enhance readability, we present the equivalence here as well.

Let $\mathcal{U} \triangleq \mathcal{X}^{|\mathcal{S}|}$, where each $u \in \mathcal{U}$ is referred to as a decision rule because the $i$-th element of $u$, denoted by $u(i)$, can be viewed as an action for state $i$. A control policy is thus a collection of decision rules spanning the entire time horizon. To facilitate the capacity analysis, we convert the *action-state channel* into an extended action-state (EAS) channel, as depicted in Fig. 6, using Shannon's method (Shannon, 1958). In particular, we conceptually separate the encoder and controller, considering the controller as an integral component of the EAS channel. We then assume that the channel state is available at the controller but not the encoder. At each time $t$, the encoder selects a decision rule $u_t$ from $\mathcal{U}$. Then the controller uses $u_t$ and the state $s_t$ to determine an action $x_t = u_t(s_t)$. Consequently, the EAS channel has an input alphabet $\mathcal{U}$, output alphabet $\mathcal{S}$, and channel law:

$$P_{S^+|S,U}(s_{t+1}|s_t, u_t) = P_{S^+|S,X}(s_{t+1}|s_t, u_t(s_t)) = \boldsymbol{T}(s_{t+1}|s_t, u_t(s_t)).$$

The EAS channel and the *action-state channel* are equivalent, and we will examine the EAS channel instead of the *action-state channel* to derive the capacity.

Assume that the initial state of the *action-state channel* is fixed to be $s_1$. Then it is well-known that the capacity of the EAS channel is (Gallager, 1968)

$$C(s_1) = \max_{\{p(u_i|u^{i-1})\}_{i \geq 1}} \lim_{N \to \infty} \frac{1}{N} I(U^N; S_2^{N+1}|s_1), \tag{14}$$

where $I(U^N; S_2^{N+1}|s_1)$ is the mutual information given by

$$I(U^N; S_2^{N+1}|s_1) = \sum_{u^N \in \mathcal{U}^N} \sum_{s_2^{N+1} \in \mathcal{S}^N} p(u^N) P(s_2^{N+1}|u^N, s_1) \log \frac{P(s_2^{N+1}|u^N, s_1)}{\sum_{z^N \in \mathcal{U}^N} p(z^N) P(s_2^{N+1}|z^N, s_1)}.$$

For two random variables $S$ and $X$, let $H(X|S)$ denote the conditional entropy (Cover, 1999) of $X$ given $S$. Then we have

$$\lim_{N \to \infty} \frac{1}{N} I(U^N; S_2^{N+1}|s_1) = \lim_{N \to \infty} \frac{1}{N} \sum_{i=1}^{N} \left[ H(S_{i+1}|S_2^i, s_1) - H(S_{i+1}|S_2^i, U^N, s_1) \right]$$

$$\overset{(a)}{=} \lim_{N \to \infty} \frac{1}{N} \sum_{i=1}^{N} \left[ H(S_{i+1}|S_2^i, s_1) - H(S_{i+1}|S_2^i, U^N, X_i, s_1) \right]$$

$$= \lim_{N \to \infty} \frac{1}{N} \sum_{i=1}^{N} \left[ H(S_{i+1}|S_i, s_1) - H(S_{i+1}|S_i, X_i, s_1) \right]$$

$$= \lim_{N \to \infty} \frac{1}{N} \sum_{i=1}^{N} I(X_i; S_{i+1}|S_i, s_1)$$

where (a) holds because $X_i$ is determined by $S^i$ and $U_i$. It follows that

$$C = \max_{\{p(u_i|u^{i-1})\}_{i \geq 1}} \lim_{N \to \infty} \frac{1}{N} I(U^N; S_2^{N+1}|s_1)$$

$$\stackrel{(a)}{=} \max_{\{p(x_i|s^i, x^{i-1})\}_{i \geq 1}} \lim_{N \to \infty} \frac{1}{N} \sum_{i=1}^{N} I(X_i; S_{i+1}|S_i, s_1)$$

$$\stackrel{(b)}{=} \max_{\{p(x_i|s_i)\}_{i \geq 1}} \lim_{N \to \infty} \frac{1}{N} \sum_{i=1}^{N} I(X_i; S_{i+1}|S_i, s_1) \tag{15}$$

In equation 15, (a) follows from the equivalence between the *action-state channel* and the EAS channel. We next prove (b). Let $\eta$ denote a history-dependent encoder and $\{\eta_i(x_i|s^i, x^{i-1})\}_{i \geq 1}$ denote the associated input distribution. Furthermore, let $P_i^\eta(s, x)$ denote the probability that $S_i = s$ and $X_i = s$ conditioned on the encoder $\eta$. Note that $\eta$ can be viewed as a history-dependent policy for the MDP. Then according to MDP theory, for any history-dependent policy $\eta$, there exists a Markov policy $\eta'$ such that $\eta$ and $\eta'$ share the same joint probability distribution of states and actions. In particular, denote by $\eta_i'(x_i|s_i)$ the probability of selecting action $x_i$ given that the state is $s_i$ at time $i$. Then $\eta'$ can be seen as a Markov encoder with input distribution $\{\eta_i'(x_i|s_i)\}_{i \geq 1}$. By letting

$$\eta_i'(x_i|s_i) = \eta_i(x_i|s_i) \triangleq \sum_{s^{i-1}, x^{i-1}} \eta_i(x_i|s^i, x^{i-1}) P(s^{i-1}, x^{i-1}|s_1),$$

we have

$$P_i^\eta(s, x|s_1) = P_i^{\eta'}(s, x|s_1), \ \forall i \geq 1, s \in \mathcal{S}, x \in \mathcal{X}$$

We omit the proof here. The interested readers are referred to Theorem 5.5.1 of (Puterman, 2014) for the formal statement and proof. Consequently, we can verify that for any history-dependent encoder $\eta$, there exists a Markov encoder $\eta'$ such that they result in the same $I(X_i; S_{i+1}|S_i, s_1)$ for all $i$:

$$I_\eta(X_i; S_{i+1}|S_i, s_1) = \sum_{s_i \in \mathcal{S}} \sum_{x_i \in \mathcal{X}} P_i^\eta(s_i, x_i|s_1) \sum_{s_{i+1} \in \mathcal{S}} \boldsymbol{T}(s_{i+1}|s_i, x_i) \log \frac{\boldsymbol{T}(s_{i+1}|s_i, x_i)}{\sum_{x_i'} \boldsymbol{T}(s_{i+1}|s_i, x_i') \eta_i(x_i'|s_i)}$$

$$= \sum_{s_i \in \mathcal{S}} \sum_{x_i \in \mathcal{X}} P_i^{\eta'}(s_i, x_i|s_1) \sum_{s_{i+1} \in \mathcal{S}} \boldsymbol{T}(s_{i+1}|s_i, x_i) \log \frac{\boldsymbol{T}(s_{i+1}|s_i, x_i)}{\sum_{x_i'} \boldsymbol{T}(s_{i+1}|s_i, x_i') \eta_i'(x_i'|s_i)}$$

We thus conclude that restricting on Markov encoders does not result in any loss of capacity.

Next, we show that, as far as finding a capacity-achieving encoder is concerned, it is enough to consider stationary encoders. To see this, we reformulate equation 15 as a dynamic programming (DP) defined as follows:

- state at time $i$: $s_i \in \mathcal{S}$
- action at time $i$ given state $s_i$: $q_i(s_i) \in \Delta(\mathcal{X})$ with $q_i(x_i|s_i) = P(x_i|s_i)$ being the $i$-th element
- transition law: $P(s_{i+1}|s_i, x_i) = \sum_{x_i \in \mathcal{X}} q_i(x_i|s_i) \boldsymbol{T}(s_{i+1}|s_i, x_i)$
- reward function:

$$r(s_i, q_i(s_i)) = \sum_{x_i \in \mathcal{X}} q_i(x_i|s_i) \sum_{s_{i+1} \in \mathcal{S}} \boldsymbol{T}(s_{i+1}|s_i, x_i) \log \frac{\boldsymbol{T}(s_{i+1}|s_i, x_i)}{\sum_{x_i'} \boldsymbol{T}(s_{i+1}|s_i, x_i') q_i(x_i'|s_i)}.$$

Then the capacity expression given in equation 15 can be written as

$$C = \max_{\{q_i\}_{i \geq 1}} \lim_{N \to \infty} \frac{1}{N} \sum_{i=1}^{N} \sum_{s_i \in \mathcal{S}} P(s_i) r(s_i, q_i(s_i)). \tag{16}$$

We can think of equation 16 as a problem of maximizing the long-term average reward of the above DP over the set of Markov deterministic policies. Essentially, the DP is an MDP with finite state space, compact action space, and bounded reward function. We can easily verify the following:

1. The reward function is a continuous function of action $d_i$.

2. The transition law depends continuously on the action $d_i$.

3. Any stationary policy yields a Markov chain with one ergodic class and a possibly empty set of transient states (by our assumption).

Then, according to MDP theory (see, e.g., (Fainberg, 1976) and (Hernández-Lerma & Lasserre, 2012)), the maximum of equation 16 can be attained by a stationary deterministic policy. Therefore, problem equation 16 is equivalent to

$$C = \max_{\{\pi(\cdot|s) \in \Delta(\mathcal{X}): s \in \mathcal{S}\}} \lim_{N \to \infty} \frac{1}{N} \sum_{i=1}^{N} I(X_i; S_{i+1}|S_i, s_1). \tag{17}$$

Note that each $\{\pi(\cdot|s) \in \Delta(\mathcal{X}) : s \in \mathcal{S}\}$ corresponds to a stationary policy for the original MDP. Under the assumption that the MDP is unichain, any stationary policy yields a Markov chain with equilibrium state distribution $\rho_\pi(s)$. Therefore, the following probability converges to a probability that is independent of $s_1$:

$$\lim_{i \to \infty} P(X_i = x, S_i = s, S_{i+1} = s'|s_1) = \rho_\pi(s)\pi(x|s)\boldsymbol{T}(s'|s, x).$$

As a result, $I(X_i; S_{i+1}|S_i, s_1)$ also converges to a value that is independent of $s_1$. The desired result follows immediately.

### B.2 PROOF OF THEOREM 2

We first show that the capacity of the *action-state channel* without reward constraint can be written as:

$$C = \max_{w \in \mathcal{W}} I(w, \boldsymbol{T}). \tag{18}$$

To see this, using Theorem 1 and the formula $\pi(x|s) = w_\pi(s, x)/\sum_{x'} w_\pi(s, x')$ yields

$$\begin{aligned}
I(X; S'|S) &= \sum_{s \in \mathcal{S}} \sum_{x \in \mathcal{X}} \rho_\pi(s)\pi(x|s) \sum_{s' \in \mathcal{S}} \boldsymbol{T}(s'|s, x) \log \frac{\boldsymbol{T}(s'|s, x)}{\sum_{x'} \boldsymbol{T}(s'|s, x')q(x'|s)} \\
&= \sum_{s \in \mathcal{S}} \sum_{x \in \mathcal{X}} w_\pi(s, x) \sum_{s' \in \mathcal{S}} \boldsymbol{T}(s'|s, x) \log \frac{\boldsymbol{T}(s'|s, x) \sum_{x''} w_q(s, x'')}{\sum_{x'} \boldsymbol{T}(s'|s, x')w_q(s, x')}.
\end{aligned}$$

Then the equivalence between equation 18 and the capacity expression given in Theorem 1 follows immediately from the one-to-one mapping between $\mathcal{W}$ and $\Pi_S$.

It is well-known that the long-term average reward of a policy can be expressed as a linear function of its occupation measure:

$$G_\pi = \lim_{N \to \infty} \frac{1}{N} \mathbb{E}_\pi \left[ \sum_{t=1}^{N} r(s_t, x_t)|s_1 \sim \alpha \right] = \sum_{s \in \mathcal{S}} \sum_{x \in \mathcal{X}} w_\pi(s, x)r(s, x). \tag{19}$$

Since the reward constraint is linear, to show that the optimization problem is a convex optimization, it suffices to prove that $I(w, \boldsymbol{T})$ is a concave function. For any $\lambda \in [0, 1]$ and $\bar{\lambda} = 1 - \lambda$, let $w = \lambda w_1 + \bar{\lambda}w_2$, where $w_1, w_2 \in \mathcal{W}$. Then we have

$$\begin{aligned}
&\sum_{x \in \mathcal{X}} w(s, x)\boldsymbol{T}(s'|s, x) \log \frac{\sum_{x'} \boldsymbol{T}(s'|s, x')w(s, x')}{\boldsymbol{T}(s'|s, x) \sum_{x''} w(s, x'')} \\
&= \sum_{x \in \mathcal{X}} w(s, x)\boldsymbol{T}(s'|s, x) \log \frac{\sum_{x'} \boldsymbol{T}(s'|s, x')w(s, x')}{\sum_{x''} w(s, x'')} - \sum_{x \in \mathcal{X}} w(s, x)\boldsymbol{T}(s'|s, x) \log \boldsymbol{T}(s'|s, x).
\end{aligned} \tag{20}$$

The second term of equation 20 is clearly linear w.r.t. $w$. For the first term, an application of log-sum inequality gives

$$
\left( \sum_{x \in \mathcal{X}} w(s,x) \boldsymbol{T}(s'|s,x) \right) \log \frac{\sum_{x'} \boldsymbol{T}(s'|s,x') w(s,x')}{\sum_{x''} w(s,x'')}
$$
$$
\leq \lambda \left( \sum_{x \in \mathcal{X}} w_1(s,x) \boldsymbol{T}(s'|s,x) \right) \log \frac{\sum_{x'} \boldsymbol{T}(s'|s,x') w_1(s,x')}{\sum_{x''} w_1(s,x'')}
$$
$$
+ \bar{\lambda} \left( \sum_{x \in \mathcal{X}} w_2(s,x) \boldsymbol{T}(s'|s,x) \right) \log \frac{\sum_{x'} \boldsymbol{T}(s'|s,x') w_2(s,x')}{\sum_{x''} w_2(s,x'')}. \tag{21}
$$

Combining equation 20 and equation 21 yields

$$
\sum_{x \in \mathcal{X}} w(s,x) \boldsymbol{T}(s'|s,x) \log \frac{\boldsymbol{T}(s'|s,x) \sum_{x''} w(s,x'')}{\sum_{x'} \boldsymbol{T}(s'|s,x') w(s,x')}
$$
$$
\geq \lambda \sum_{x \in \mathcal{X}} w_1(s,x) \boldsymbol{T}(s'|s,x) \log \frac{\boldsymbol{T}(s'|s,x) \sum_{x''} w_1(s,x'')}{\sum_{x'} \boldsymbol{T}(s'|s,x') w_1(s,x')}
$$
$$
+ \bar{\lambda} \sum_{x \in \mathcal{X}} w_2(s,x) \boldsymbol{T}(s'|s,x) \log \frac{\boldsymbol{T}(s'|s,x) \sum_{x''} w_2(s,x'')}{\sum_{x'} \boldsymbol{T}(s'|s,x') w_2(s,x')}. \tag{22}
$$

Summing both sides of the above inequality over $s$ and $s'$ yields

$$
I(w, \boldsymbol{T}) = I(\lambda w_1 + \bar{\lambda} w_2, \boldsymbol{T}) \geq \lambda I(w_1, \boldsymbol{T}) + \bar{\lambda} I(w_2, \boldsymbol{T}).
$$

We thus conclude that $I(w, \boldsymbol{T})$ is a concave function of $w$. This completes the proof.

### B.3 PROOF OF LEMMA 1

Define

$$
\mathcal{W}_V = \left\{ w \in \mathcal{W} : \sum_{s \in \mathcal{S}} \sum_{x \in \mathcal{X}} w(s,x) r(s,x) \geq V \right\}.
$$

For any achievable $V_1$ and $V_2$, let

$$
w_i = \arg \max_{w \in \mathcal{W}_{V_i}} I(w, \boldsymbol{T}), \ i = 1, 2.
$$

Then $C(V_i) = I(w_i, \boldsymbol{T})$, $i = 1, 2$. For any $\theta \in [0, 1]$, let $\bar{\theta} = 1 - \theta$. Let $V = \theta V_1 + \bar{\theta} V_2$ and $w_3 = \theta w_1 + \bar{\theta} w_2$. Then clearly $w_3 \in \mathcal{W}_V$. We thus have

$$
C(V) = \max_{w \in \mathcal{W}_V} I(w, \boldsymbol{T}) \geq I(w_3, \boldsymbol{T}) \geq \theta I(w_1, \boldsymbol{T}) + \bar{\theta} I(w_2, \boldsymbol{T}) = \theta C(V_1) + \bar{\theta} C(V_2).
$$

We thus conclude that $C(V)$ is concave.

### B.4 PROOF OF LEMMA 2

Since $I(w, \boldsymbol{T})$ is concave w.r.t. $w$ and $l(w, w_n, \boldsymbol{T})$ is linear w.r.t. $w$, to show that $l(w, w_n, \boldsymbol{T})$ is a tangent line of $I(w, \boldsymbol{T})$ at point $w_n$, it is enough to prove that statements (i) and (ii) hold. Statement (i) holds trivially by the definitions of the two functions.

For statement (ii), consider

$$
l(w, w_n, \boldsymbol{T}) - I(w, \boldsymbol{T})
$$
$$
= \sum_{s \in \mathcal{S}} \sum_{x \in \mathcal{X}} w(s,x) \sum_{s' \in \mathcal{S}} \boldsymbol{T}(s'|s,x) \log \frac{\sum_{x''} w_n(s,x'')}{\sum_{x'} \boldsymbol{T}(s'|s,x') w_n(s,x')} \frac{\sum_{x'} \boldsymbol{T}(s'|s,x') w(s,x')}{\sum_{x''} w(s,x'')}.
$$

Define

$$
P_w(s'|s) = \frac{\sum_{x'} \boldsymbol{T}(s'|s,x') w(s,x')}{\sum_{x''} w(s,x'')}, \ P_{w_n}(s'|s) = \frac{\sum_{x'} \boldsymbol{T}(s'|s,x') w_n(s,x')}{\sum_{x''} w_n(s,x'')}.
$$

Note that $\sum_{s'} P_w(s'|s) = 1$. Hence it is indeed a conditional probability. Then

$$
\begin{aligned}
l(w, w_n, \boldsymbol{T}) - I(w, \boldsymbol{T}) &= \sum_{s \in \mathcal{S}} \sum_{s' \in \mathcal{S}} \left( \sum_{x \in \mathcal{X}} w(s, x) \boldsymbol{T}(s'|s, x) \right) \log \frac{P_w(s'|s)}{P_{w_n}(s'|s)} \\
&= \sum_{s \in \mathcal{S}} \sum_{x \in \mathcal{X}} w(s, x) \sum_{s' \in \mathcal{S}} P_w(s'|s) \log \frac{P_w(s'|s)}{P_{w_n}(s'|s)} \\
&\geq 0,
\end{aligned}
$$

where the inequality follows from the non-negativity of relative entropy. Therefore, $l(w, w_n, \boldsymbol{T})$ is a tangent line of $I(w, \boldsymbol{T})$ at point $w_n$.

## C SYSTEM EXPLANATION AND ALGORITHM DETAILS

In this Appendix, we first clarify the relationship between the theoretical results in Section 4 and the algorithm in Section 5, as this distinction may be unclear to those unfamiliar with information and coding theory. Following this clarification, we provide additional details about the components of *Act2Comm*.

The concave optimization problem in Theorem 2 characterizes the capacity-reward tradeoff of the action-state channel. It provides an upper bound for practically achievable coding rates under a certain reward constraint. Solving the concave optimization yields the capacity and the optimal state-action distribution $\omega$, which can be translated to a stationary policy for the MDP via the following formula:

$$
\pi(a|s) = \frac{\omega(s, a)}{\sum_{a'} \omega(s, a')}.
$$

Note that this policy $\pi$ can not be directly used as the coding policy for communication, as the coding policy needs to generate actions based on both the state and the message. Therefore, in Section 5, we propose *Act2Comm* to learn a coding policy that mimics the behavior of policy $\pi$ from the perspective of MDP control, with the input of message and feedback block.

For practical channel coding in the finite block-length regime, historical feedback has proven beneficial based on prior experience with conventional channel coding. To effectively map messages and feedback to actions, the transformer architecture with an attention mechanism is naturally well-suited, forming the backbone for both the encoder and decoder components.

The components of communication via actions in MDPs using the proposed *Act2Comm* are summarized in Table. 2. In particular,

- The encoder encodes a belief matrix $\boldsymbol{Z}^{(\tau)} \in \mathbb{R}^{\tau \times \frac{\mu|\mathcal{S}|}{R}}$ using message block $\boldsymbol{B}^{(\tau)}$ and feedback block $\boldsymbol{C}^{(\tau)}$ at each coding round $\tau$. After completing all $l$ coding rounds, a final belief map $\boldsymbol{Z} \in \mathbb{R}^{\frac{k}{\mu} \times \frac{\mu|\mathcal{S}|}{R}}$ is constructed. Note that instead of directly outputting actions at each time step, our encoder generates a continuous belief map, which is subsequently quantized into decision rules. This approach not only enhances performance but also simplifies the training of the critic network.

- A quantizer then generates the codeword $\boldsymbol{U} \in \mathcal{X}^{|\mathcal{S}| \times \frac{k}{R}}$, where each element of the resultant codeword, $x_t = \boldsymbol{U}[s_t, t]$, represents the selected action for state $s_t$ at time step $t$.

- Given the action and state, EAS channel then returns the next state.

- The decoder collects all states and then performs the decoding process, which outputs logits $\hat{\boldsymbol{M}} \in \mathbb{R}^{\frac{k}{\mu} \times 2^\mu}$ for the message.

- Since the gradient cannot propagate through the EAS and quantizer to the encoder, a critic network is introduced to link the belief map $\boldsymbol{Z}$ to the decoded logits $\hat{\boldsymbol{M}}$. Specifically, the critic network is trained to predict $\hat{\boldsymbol{M}}_k$ from $\boldsymbol{Z}_k$, producing $\hat{\boldsymbol{M}}_{ck}$ at each inner optimization step $k$ within the total $s_{in}$ steps. Thanks to the introduction of the critic network, the gradient can propagate through it, from logits to the belief map. As shown in the table,

Table 2: Functionality of Each System Component

| Model components | Function | Input | Output |
|---|---|---|---|
| Encoder | Coding the belief matrix | $B^{(\tau)}, C^{(\tau)}$ | $Z$ |
| Quantizer | Generate decision rules | $Z$ | $U$ |
| EAS channel | Generate state from actions | $U \to x_t$ | $s_{t+1}$ |
| Decoder | Decoding the message | $s$ | $\hat{M} \to \hat{m}$ |
| Critic Network | Estimate decoding logits | $Z_k$ | $\hat{M}_{ck}$ |

it effectively "links" the encoder's output to the decoder's output, thereby "skipping" the quantizer, EAS channel and decoder, which are treated as the unknown environment during the training phase.

## C.1 QUANTIZER

The quantizer $\mathcal{Q} : \mathbb{R}^{|\mathcal{S}| \times \frac{k}{R}} \to \mathcal{X}^{|\mathcal{S}| \times \frac{k}{R}}$ is designed to convert real-valued coding results $|\mathcal{X}| \cdot \text{Sigmoid}(Z)$ into integers corresponding to actions within the channel input alphabet set $\mathcal{X} = \{0, 1, \ldots, |\mathcal{X}| - 1\}$. For example, if there exist 5 actions and 1 state, then each element of the coding result $|\mathcal{X}| \cdot \text{Sigmoid}(Z) \in (0, 5)$ will be rounded down into the nearest action index. If $|\mathcal{X}| \cdot \text{Sigmoid}(Z) = [1.5, 0.8, 2.1, 3.2, 4.8]$, it will be rounded down to $[1, 0, 2, 3, 4]$ as the channel input via the quantizer $\mathcal{Q}$.

Here, we adopt hard rounding, despite the availability of advanced quantization methods such as soft rounding or noise injection during training. The chosen approach is intentionally straightforward, and a critic network effectively mitigates the non-differentiability introduced by hard rounding.

## C.2 CRITIC NETWORK AND ITERATIVE TRAINING

As shown in Fig. 7, a critic network is introduced to facilitate gradient backpropagation for encoder optimization, treating the quantizer, channel, and decoder as an unknown environment. Before updating the encoder and decoder, the Critic network is trained over $s_{in}$ steps to capture knowledge of the environment, with the encoder and decoder frozen during this phase. Subsequently, the encoder is updated with the assistance of the Critic network, aiming to jointly optimize for control and communication, while keeping the decoder frozen. Following this, the optimized encoder is frozen, and the decoder is updated based on the newly optimized encoder. With the updated decoder, the Critic network is retrained to adapt to the updated environment, preparing it to support the next round of encoder updates.

This process iteratively alternates between freezing one component while optimizing the other. From the experiment, we observe that the training process is more stable when the encoder is updated once for every two updates of the decoder. The effectiveness of this approach is visualized in a Fig. 11 and Table 5. The detailed training algorithm is presented in the Algorithm. 7

## C.3 TRAINING AND INFERENCE ALGORITHM

To enhance readers' understanding of the training and inference process, we provide the pseudocode and illustration figures for (see Fig. 7) Act2Comm, detailing both the training and inference phases, as shown in Algorithms 1 and 2. Furthermore, Fig. 11 visualizes the training process by tracking the loss values throughout the iterative updates. For additional details about the training, the training logs and source code are also available on the project page of this paper.

To be more specific for the training phase, we train the encoder first and then train the decoder. The critic network, with control loss, is applied to the encoder. The decoder only considers the communication loss. After training, the critic network is removed, as shown in Fig. 7 (c). Only the encoder and decoder are deployed for communication and control, as shown in Fig. 8b.

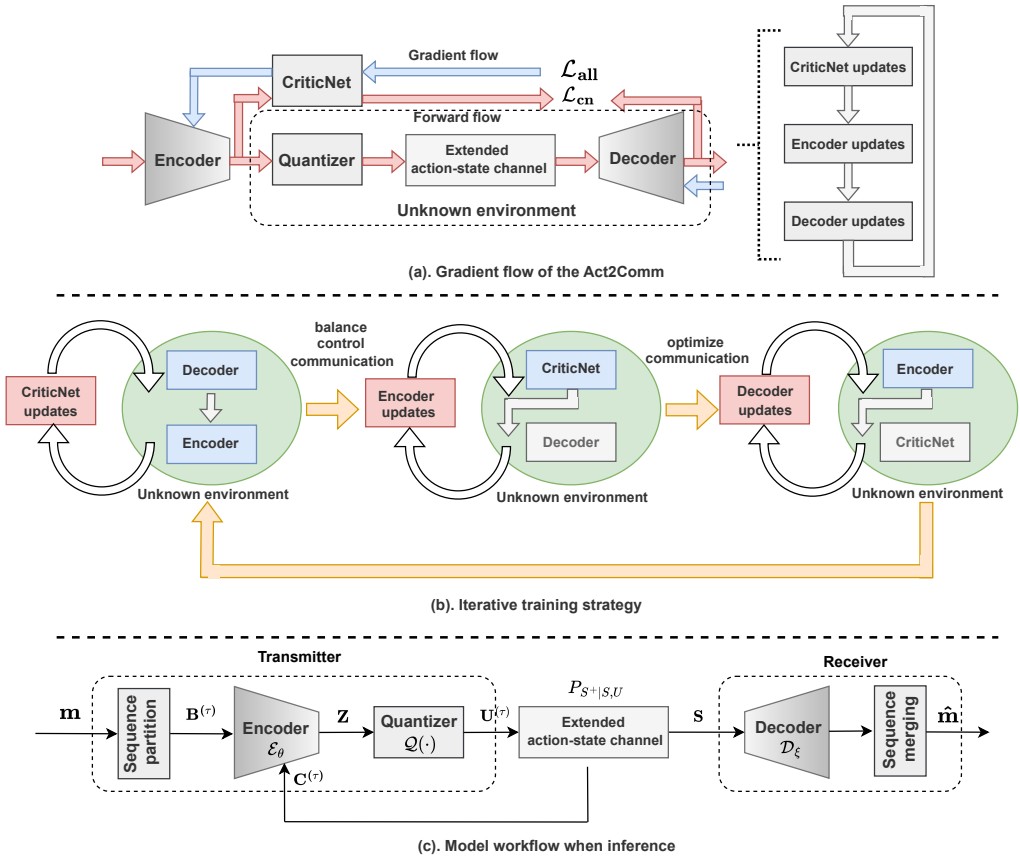

Figure 7: Illustration of the iterative training process: (a) Gradient flow in the proposed method, where blue arrows indicate the gradient flow, and red arrows represent the forward process. (b) Diagram of the update steps, where the red block represents the component being updated, the blue block represents the frozen component, and the grey block indicates an unused component. (c) Inference phase for the well-trained *Act2Comm* model, where the critic network is removed.

Table 3: Number of parameters, FLOPs, and coding times for the *Act2Comm* scheme ($K = 12$, $\mu = 3$, $R = 1/3$, $s_{in} = 20$) with varying state numbers and action numbers, where the increased value are colored with orange and red.

| Action and state number | Encoder | | | Decoder | | |
|:---:|:---:|:---:|:---:|:---:|:---:|:---:|
| $(|\mathcal{A}|, |\mathcal{S}|)$ with $|\mathcal{A}| \uparrow$ | $(5, 16)$ | $(20, 16)$ | $(40, 16)$ | $(5, 16)$ | $(20, 16)$ | $(40, 16)$ |
| **Parameters (k)** | 32.244 | 32.244 | 32.244 | 50.536 | 50.536 | 50.536 |
| **FLOPs (millions)** | 0.3164 | 0.3164 | 0.3164 | 0.1989 | 0.1989 | 0.1989 |
| **Coding Time (ms)** | 6.691 | 6.691 | 6.691 | 2.613 | 2.613 | 2.613 |
| $(|\mathcal{A}|, |\mathcal{S}|)$ with $|\mathcal{S}| \uparrow$ | $(5, 16)$ | $(5, 64)$ | $(5, 256)$ | $(5, 16)$ | $(5, 64)$ | $(5, 256)$ |
| **Parameters (k)** | 32.244 | 46.51 | 103.52 | 50.536 | 50.536 | 50.536 |
| **FLOPs (millions)** | 0.3164 | 0.4546 | 1.007 | 0.1989 | 0.1989 | 0.1989 |
| **Coding Time (ms)** | 6.691 | 6.69 | 7.04 | 2.613 | 2.613 | 2.613 |

## C.4 COMPLEXITY ANALYSIS

To analyze the complexity of the proposed method, we consider an input size of 12 bits, 4 blocks, and a rate of $R = 1/3$. The experimental results, presented in Table 4, were obtained using a single GPU-A5000 with $10,000$ runs for the "Erratic Robot" environment.

From the table, we observe that during the training process, the main computational complexity arises from the critic network, as it needs to run around 20 iterations for each encoder update. Note

---

**Algorithm 1** Training strategy of the *Act2Comm* scheme

---

**Input:** The initial MDP state: $s_0$, Inner optimization step: $s_{in}$,
Learning rates of encoder, decoder, and critic network: $\beta, \alpha, \alpha$, with $\alpha = 5\beta$,
Target policy: $\boldsymbol{\pi}$, balanced parameter $\lambda$.
**Output:** Parameters of the encoder and decoder: $\boldsymbol{\theta}, \boldsymbol{\xi}$.

---

1: **for** the $i$-th step within the total training steps **do**
2:    $\mathbf{m} = [\boldsymbol{b_1}; \boldsymbol{b_2}; \ldots; \boldsymbol{b_l}] \in \{0,1\}^{l \times \mu}$,        *// Randomly sample and divide the message*
3:    $\mathbf{C}^{(1)} \in 0^{1 \times \frac{\mu}{R} \times 3}$,             *// Initialize the feedback block*
4:    **for** each coding round $\tau$ $(1 \leq \tau \leq l)$ **do**
5:       $\mathbf{B}^{(\tau)} = [2\boldsymbol{b_1} - 1; \ldots; 2\boldsymbol{b_\tau} - 1] \in \mathbb{R}^{\tau \times \mu}$,    *// Construct the message block*
6:       $\mathbf{C}^{(\tau)} = [\boldsymbol{C_1}; \ldots; \boldsymbol{C_\tau}] \in \mathbb{R}^{\tau \times \frac{\mu}{R} \times 3}$,    *// Update the feedback block*
7:       $\mathbf{Z}^{(\tau)} = \mathcal{E}_{\boldsymbol{\theta}}(\mathbf{B}^{(\tau)}, \mathbf{C}^{(\tau)}) \in \mathbb{R}^{\tau \times \frac{\mu|S|}{R}}$,
      $\mathbf{z}^{(\tau)} = \boldsymbol{Z^{(\tau)}}[\tau, :] \in \mathbb{R}^{\frac{\mu|S|}{R}} \to \mathbb{R}^{|S| \times \frac{\mu}{R}}$,    *// Encoding the belief map*
8:       $\mathbf{U}^{(\tau)} = \mathcal{Q}(|\mathcal{X}|\text{Sigmoid}(\boldsymbol{z^{(\tau)}})) \in \mathcal{X}^{|S| \times \frac{\mu}{R}}$,    *// Quantization for decision rules*
9:       **for** $t = 1 : 1 : \frac{\mu}{R}$ **do**
10:          $s_{t+1} = \boldsymbol{T}(s_{t+1}|s_t, \mathbf{U}^{(\tau)}[s_t, t])$,    *// Go through the EAS channel for $\frac{\mu}{R}$ time steps*
11:          $\boldsymbol{c_t^{(\tau)}} = [s_t^{(\tau)}, x_t^{(\tau)}, s_{t+1}^{(\tau)}] \in \mathbb{R}^{1 \times 3}$,    *// Update the feedback vector*
12:       **end for**
13:       $\boldsymbol{C_\tau} = [\boldsymbol{c_1^{(\tau)}}; \ldots; \boldsymbol{c_{\frac{\mu}{R}}^{(\tau)}}] \in \mathbb{R}^{\frac{\mu}{R} \times 3}$,    *// Update the feedback matrix*
14:    **end for**
15:    $\boldsymbol{Z} = [\boldsymbol{z^{(1)}}; \ldots; \boldsymbol{z^{(l)}}] \in \mathbb{R}^{\frac{k}{\mu} \times \frac{\mu|S|}{R}}$, $\boldsymbol{s} = [s_1; \ldots; s_{\frac{k}{R}}] \in \mathcal{S}^{\frac{k}{R}} \to \mathcal{S}^{\frac{k}{\mu} \times \frac{\mu}{R}}$,
16:    $\hat{M} = \mathcal{D}(\boldsymbol{s}) \in \mathbb{R}^{\frac{k}{\mu} \times 2^\mu}$,    *// Collect codewords, states, and decode the logits*
17:    ——————————————————— *// Train the Critic network*
18:    **if** $i\%2 == 0$ **then**
19:       **for** $k = 1 : 1 : s_{in}$ **do**
20:          $\boldsymbol{Z_k} = \boldsymbol{Z} + \boldsymbol{W_k}$, with $\boldsymbol{W_k} \in \mathbb{R}^{\frac{k}{\mu} \times \frac{\mu|S|}{R}} \sim \mathcal{N}(0, \sigma_w^2)$    *// Neighboring sampling*
21:          $\mathbf{U_k} = \mathcal{Q}(|\mathcal{X}|\text{Sigmoid}(\boldsymbol{Z_k})) \in \mathcal{X}^{|S| \times \frac{k}{R}}$,
22:          **for** $t = 1 : 1 : \frac{k}{R}$ **do**
23:             $s_{t+1} = \boldsymbol{T}(s_{t+1}|s_t, \mathbf{U}_{(\mathbf{k})}[s_t, t])$,    *// Go through the EAS channel*
24:          **end for**
25:          $\hat{\boldsymbol{s}}_{(\boldsymbol{k})} = [s_1; \ldots; s_{\frac{k}{R}}] \in \mathcal{S}^{\frac{k}{R}} \to \mathcal{S}^{\frac{k}{\mu} \times \frac{\mu}{R}}$,    *// Collect the observed states*
26:          $\hat{\boldsymbol{M}}_{\boldsymbol{k}} = \mathcal{D}(\hat{\boldsymbol{s}}_{(\boldsymbol{k})}) \in \mathbb{R}^{\frac{k}{\mu} \times 2^\mu}$,    *// Decode the logits with a frozen decoder*
27:          $\hat{\boldsymbol{M}}_{\boldsymbol{ck}} = \mathcal{C}(\boldsymbol{Z_k}) \in \mathbb{R}^{\frac{k}{\mu} \times 2^\mu}$    *// Predict the logits with critic network*
28:          $\mathcal{L}_{cn} = \text{MSE}(\hat{\boldsymbol{M}}_{\boldsymbol{ck}}, \hat{\boldsymbol{M}}_{\boldsymbol{k}})$    *// Compute the critic loss with MSE*
29:          $\phi_{\boldsymbol{k+1}} = \phi_{\boldsymbol{k}} - \beta\nabla_\phi\mathcal{L}_{cn}$    *// Update the Critic network.*
30:       **end for**
31:    ——————————————————— *// Train the Encoder with a frozen Decoder*
32:       $\mathcal{L}_{cont} = \text{MSE}(\boldsymbol{\pi}, \hat{\boldsymbol{f}})$    *// Control loss term for the estimated $\hat{f}$*
33:       $\mathcal{L}_{com} = \text{cross-entropy}(\boldsymbol{m}, \hat{\boldsymbol{M}}_{\boldsymbol{ck}})$    *// Communication loss term*
34:       $\boldsymbol{\theta} = \boldsymbol{\theta} - \alpha\nabla_{\boldsymbol{\theta}}(\mathcal{L}_{com} + \lambda\mathcal{L}_{cont})$    *// Update the encoder*
35:    **end if**
36:    ——————————————————— *// Train the Decoder with a frozen Encoder*
37:    **for** each coding round and corresponding time step **do**
38:       $\mathbf{z}^{(\tau)} = \mathcal{E}_{\boldsymbol{\theta}}(\mathbf{B}^{(\tau)}, \mathbf{C}^{(\tau)})[\tau, :] \to \boldsymbol{U^{(\tau)}}$,    *// Re-encode and re-quantize the decision rules*
39:       $s_{t+1} = \boldsymbol{T}(s_{t+1}|s_t, \mathbf{U}^{(\tau)}[s_t, t])$,    *// Re-go through the EAS channel for all time steps*
40:    **end for**
41:    $\boldsymbol{s} = [s_1; \ldots; s_{\frac{k}{R}}]$, $\hat{\boldsymbol{M}} = \mathcal{D}(\boldsymbol{s})$    *// Decode the observed states from a forzen encoder*
42:    $\mathcal{L}_{com} = \text{cross-entropy}(\boldsymbol{m}, \hat{\boldsymbol{M}})$    *// Communication loss term*
43:    $\boldsymbol{\xi} = \boldsymbol{\xi} - \beta\nabla_{\boldsymbol{\xi}}\mathcal{L}_{com}$    *// Update the decoder*
44: **end for**

---

---

**Algorithm 2** Inference of the *Act2Comm* scheme

---

**Input and output for the encoding:**
**Input:** Initial state: $s_0$, $k$-bits message $\mathbf{m}$, coding rate $R$, well-trained encoder: $\boldsymbol{\theta}^*$;
**Output:** Codeword (decision rules) $\boldsymbol{Z} \in \mathbb{R}^{|S| \times \frac{k}{R}}$.

---

**Input and output for the decoding:**
**Input:** Observed states $\boldsymbol{s} \in \mathcal{S}^{\frac{k}{R}}$, well-trained decoder $\boldsymbol{\xi}^*$;
**Output:** Predicted message: $\hat{\boldsymbol{m}}$.

---

1: **Encoding phase:**
2: $\mathbf{C}^{(1)} \in 0^{1 \times \frac{\mu}{R} \times 3}$,                  // *Initialize the feedback block*
3: **for** each coding round $\tau$ $(1 \le \tau \le l)$ **do**
4:     $\mathbf{B}^{(\tau)} = [2\boldsymbol{b_1} - 1; \ldots; 2\boldsymbol{b_\tau} - 1] \in \mathbb{R}^{\tau \times \mu}$,                  // *Construct the message block*
5:     $\mathbf{C}^{(\tau)} = [\boldsymbol{C_1}; \ldots; \boldsymbol{C_\tau}] \in \mathbb{R}^{\tau \times \frac{\mu}{R} \times 3}$,                  // *Update the feedback block*
6:     $\mathbf{Z}^{(\tau)} = \mathcal{E}_{\boldsymbol{\theta}}(\mathbf{B}^{(\tau)}, \mathbf{C}^{(\tau)}) \in \mathbb{R}^{\tau \times \frac{\mu|S|}{R}}$,
        $\boldsymbol{z}^{(\tau)} = \boldsymbol{Z}^{(\tau)}[\tau, :] \in \mathbb{R}^{\frac{\mu|S|}{R}} \to \mathbb{R}^{|S| \times \frac{\mu}{R}}$,                  // *Encoding the belief map*
7:     $\mathbf{U}^{(\tau)} = \mathcal{Q}(|\mathcal{X}|\mathrm{Sigmoid}(\boldsymbol{Z})) \in \mathcal{X}^{|S| \times \frac{\mu}{R}}$,                  // *Quantization for decision rules*
8:     **for** $t = 1 : 1 : \frac{\mu}{R}$ **do**
9:         $x_t^{(\tau)} = U^{(\tau)}[s_t, t])$
            $s_{t+1}^{(\tau)} = \boldsymbol{T}(s_{t+1}^{(\tau)}|s_t^{(\tau)}, x_t^{(\tau)})$,                  // *Go through the EAS channel for $\frac{\mu}{R}$ time steps*
10:         $\boldsymbol{c_t^{(\tau)}} = [s_t^{(\tau)}, x_t^{(\tau)}, s_{t+1}^{(\tau)}] \in \mathbb{R}^{1 \times 3}$,                  // *Update the feedback vector*
11:     **end for**
12:     $C_\tau = [\boldsymbol{c_1^{(\tau)}}; \ldots; \boldsymbol{c_{\frac{\mu}{R}}^{(\tau)}}] \in \mathbb{R}^{\frac{\mu}{R} \times 3}$,                  // *Update the feedback matrix*
13: **end for**
14: $\boldsymbol{Z} = [\boldsymbol{z^{(1)}}; \ldots; \boldsymbol{z^{(l)}}] \in \mathbb{R}^{\frac{k}{\mu} \times \frac{\mu|S|}{R}} \to \mathbb{R}^{|S| \times \frac{k}{R}}$,                  // *Collect the final codeword*

---

15: **Decoding phase:**
16: $\boldsymbol{s} = [s_1; \ldots; s_{\frac{k}{R}}] \in \mathcal{S}^{\frac{k}{R}} \to \mathcal{S}^{\frac{k}{\mu} \times \frac{\mu}{R}}$,                  // *Collect the observed states*
17: $\hat{\boldsymbol{M}} = \mathcal{D}(\boldsymbol{s}) \in \mathbb{R}^{\frac{k}{\mu} \times 2^\mu}$,                  // *Decode the logits*
18: $\hat{\boldsymbol{m}} = \mathrm{argmax}(\mathrm{Softmax}(\hat{\boldsymbol{M}}), \mathrm{dim} = -1)$                  // *Decode each block labels*

---

Table 4: Number of parameters, FLOPs, and coding times for the *Act2Comm* scheme ($K = 12$, $\mu = 3$, $R = 1/3$, $s_{in} = 20$) with varying critic network training steps $s_{in}$ for both training and inference phases. Note that the backpropagation computational complexity is approximately 2–3 times that of the forward computation.

| Training Phase (forward) | | | | | |
|---|---|---|---|---|---|
| **Model components** | **Encoder** | **Decoder** | **CrititNet** $s_{in} = 5$ | **CriticNet** $s_{in} = 10$ | **CriticNet** $s_{in} = 20$ |
| **Parameters (k)** | 32.244 | 50.536 | 7.568 | 7.568 | 7.568 |
| **FLOPs (millions)** | 0.3164 | 0.1989 | 0.1454 | 0.2909 | 0.5818 |
| **Inference Phase** | | | | | |
| **Model components** | **Encoder** | | **Decoder** | | **Critic network** |
| **Parameters (k)** | 32.244 | | 50.536 | | ✗ |
| **FLOPs (millions)** | 0.3164 | | 0.1989 | | ✗ |
| **Coding Time (ms)** | 6.691 | | 2.613 | | ✗ |

that the back-propagation FLOPs can be estimated by multiplying the forward FLOPs with a factor (typically 2–3x). During inference, the critic network is removed. We observe that the encoding process can be completed within 10 ms, while the decoding process is even faster, taking less than 3 ms for the message.

We also analyze the complexity of our approach concerning increasing state and action spaces. Specifically, based on the design of Act2Comm, the number of actions does not impact the scaling performance of the model, in terms of the computational complexity. A larger action space primarily contributes to more diversity in decision rules, without increasing the computational complexity of the approach. For increased state spaces, the complexity grows only in the encoder component. This occurs because the computational complexity of the encoder increases as the output matrix size expands with the number of states.

However, environments with more complex action or state spaces may lead to a more challenging learning process, potentially affecting performance. To validate our method in such scenarios, we introduced a new environment, the "Erratic Robot," which features a five-action space. Results demonstrate that our method remains effective, even in this more complex setting.

### C.5 VECTOR REPRESENTATION OF STATES AND ACTIONS

This paper focuses on MDPs with finite state and action spaces, which allows us to represent states and actions as scalar integers by indexing them. This means that even when states and actions are vectors, they can be mapped to scalars. We adopt this approach in the proposed Act2Comm for ease of presentation. However, it is worth noting that using vector representations for states and actions can be beneficial in some MDPs. This can be implemented with minor adjustments to the feedback structure for the encoder and the input structure for the decoder.

Specifically, with vector representations, the feedback vector is defined as: $c_t^{(\tau)} \triangleq [s_t^{(\tau)}, x_t^{(\tau)}, s_{t+1}^{(\tau)}] \in \mathbb{R}^{1 \times (2n_s + n_x)}$, where $n_s$ and $n_x$ are the dimensionalities of state and action. Here, $s_t^{(\tau)} \in \mathcal{S}^{1 \times n_s}$ and $s_{t+1}^{(\tau)} \in \mathcal{S}^{1 \times n_s}$ are the vector representations of states, rather than their scalar counterparts (i.e., indices); similarly, $x_t^{(\tau)} \in \mathcal{X}^{1 \times n_x}$ is the vector representation of the action. Accordingly, the feedback matrix is defined as $C_\tau = [c_1^{(\tau)}; \ldots; c_{\frac{\mu}{R}}^{(\tau)}] \in \mathbb{R}^{\frac{\mu}{R} \times (2n_s + n_x)}$, and the feedback block can be constructed as $C^{(\tau)} \triangleq [C_1; \ldots; C_\tau] \in \mathbb{R}^{\tau \times \frac{\mu}{R} \times (2n_s + n_x)}$. Consequently, state observations $s \triangleq [s_1; \ldots; s_{\frac{k}{R}}] \in \mathcal{S}^{\frac{k}{R} \times n_s}$ are fed into the decoder to produce logits.

To evaluate the effectiveness of *Act2Comm* with vector representations, we applied this variant to the "Catch the Ball" scenario with $R = 1/3$ (cf. Fig. 4a). In this environment, the state is represented as a vector $[s_a, s_b]$, where $s_a \in \{0, 1, 2\}$ indicates the position of the board, and $s_b \in \{0, 1, 2, 3, 4, 5, 6, 7, 8\}$ indicates the position of the ball. The performance of Act2Comm with vector representations in this experiment is very similar to that achieved with scalar state representations, as shown in Fig. 4. Notably, we observe that the vector representation offers advantages such as more stable and faster convergence during training. This is illustrated in Fig. 8a, where the loss value decreases and converges more quickly during training compared to the scalar state representation counterpart. An intuitive explanation for this result is that the vector state representation captures more detailed features of the environment, which may potentially facilitate learning.

## D EXPERIMENTAL DETAILS

### D.1 EXPERIMENT SETUP

For the *Act2Comm* scheme, we train the model with a batch size of 4096, a learning rate of 0.001, and an Adam-based lookahead optimizer (Zhang et al., 2019). The inner-training for the critic network consists of $s_{in} = 20$ steps, with a noise variance of $\sigma_w^2 = 0.1$. Each block has a length of $\mu = 3$, and temperature parameter is as $\gamma = 10, \gamma = 50, \gamma = 100, \gamma = 200$. The performance presented is averaged over $20,000$ execution times. To investigate the trade-offs, we train the *Act2Comm* model with $\lambda \in [0.01, 20]$.

The detailed architecture of the *Act2Comm* scheme is provided in Fig. 8b. At the transmitter, a transformer-based encoder is utilized to generate the $z^{(\tau)}$ from $B^{(\tau)}$ and $C^{(\tau)}$ for each coding round. Specifically, $C^{(\tau)}$ and $B^{(\tau)}$ are firstly processed through multilayer perceptron (MLP) layers, then concatenated for linear projection and positional encoding operations, resulting in $F_0 \in \mathbb{R}^{\tau \times d}$, where $d$ is the hidden layer dimension. After $L_t$ transformer layers, the resultant $F_{L_t} \in \mathbb{R}^{\tau \times d}$

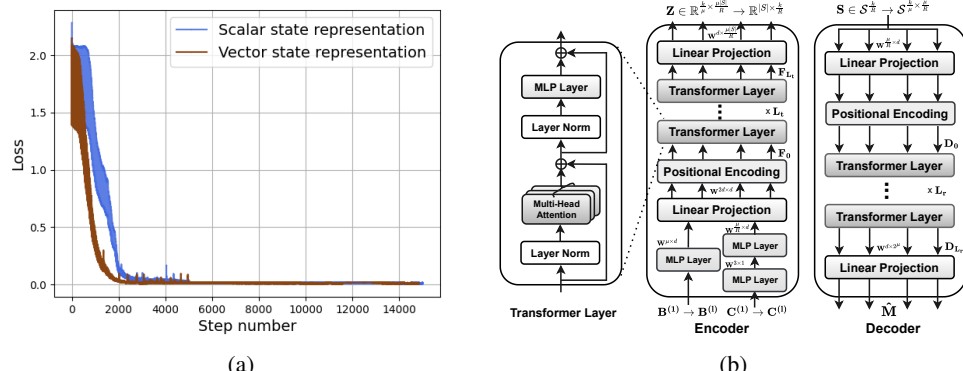

Figure 8: (a). Training process comparison for scalar state and vector state representations, where $\lambda = 20$, $\gamma = 50$ and $R = 1/3$. (b). The architecture of the *Act2Comm*.

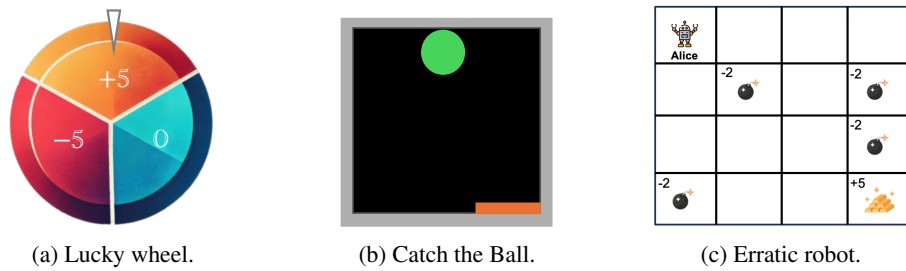

(a) Lucky wheel.  (b) Catch the Ball.  (c) Erratic robot.

Figure 9: Illustrations of experimental MDP environments

is further transformed by a fully-connected layer into $\boldsymbol{Z}^{(\tau)}$. The decoding process begins with reshaping $\boldsymbol{S}$ into $\mathcal{S}^{\frac{k}{\mu} \times \frac{\mu}{R}}$. This reshaped $\boldsymbol{S}$ is then processed symmetrically through fully-connected layers, positional encoding, and $L_r$ transformer layers, yielding the feature $\boldsymbol{D}_{\boldsymbol{L_r}} \in \mathbb{R}^{\frac{k}{\mu} \times d}$. Subsequently, $\boldsymbol{D}_{\boldsymbol{L_r}}$ is input into a fully-connected layer to generate the logits $\hat{\boldsymbol{M}} \in \mathbb{R}^{l \times 2^{\mu}}$ for each block. After a softmax function, we predict the label of $\boldsymbol{m}$ and transform it into bit stream $\hat{\boldsymbol{m}}$. Specifically, we set $d = 32$, $L_t = 2$ and $L_t = 4$ during the experiments.

## D.2 LUCKY WHEEL

As illustrated in Fig. 9a, the wheel is evenly divided into three regions. At each time step, the player can select either action $0$ or action $1$, where action $0$ corresponds to a clockwise rotation of the wheel, and action $1$ corresponds to an anti-clockwise rotation. If action $0$ is selected, there is a probability $p = 0.2$ that the pointer remains in its current region, and a probability $1 - p = 0.8$ that it moves to the next region in the clockwise direction. Similarly, if action $1$ is selected, there is a probability $p = 0.2$ that the pointer remains in its current region, and a probability $1 - p = 0.8$ that it moves to the next region in the anti-clockwise direction. The rewards for the three regions are $5$, $-5$, and $0$, respectively. The player receives a reward at each time step based on the region in which the pointer is located.

## D.3 CATCH THE BALL

The "Catch the Ball" game is set in a $3 \times 3$ grid, as illustrated in Fig. 9b. A ball randomly appears at the top of the grid and descends one grid space at each time step. Meanwhile, a board at the bottom moves horizontally to catch the falling ball. Each time the board successfully catches a ball, the player receives a reward $r$. If the ball falls to the bottom of the grid without being caught, it disappears, resulting in a penalty of $-r$ for the player. At all other times, there are no rewards or

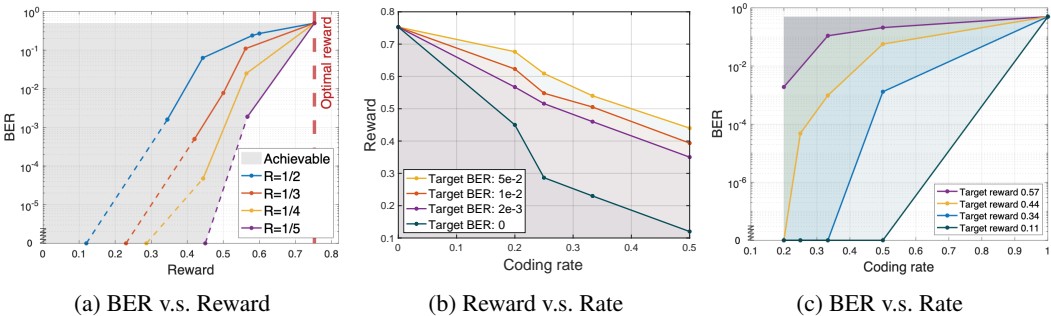

| (a) BER v.s. Reward | (b) Reward v.s. Rate | (c) BER v.s. Rate |
| --- | --- | --- |

Figure 10: Control-communication trade-off of *Act2Comm* in "Erratic Robot".

penalties for the player. After a ball disappears or is caught, a new ball appears randomly (with equal probability) at one of the three positions at the top of the grid.

The player has three available actions to move the board: move left, move right, or remain stationary. If the player chooses to move left or right, there is a probability $p \in [0, 1)$ that the movement fails (in which case the board remains in its current position), and a probability $1-p$ that the board moves for one grid space successfully as intended. Additionally, any action that attempts to move the board outside the grid will always fail. In this experiment, we set $r = 5$ and $p = 0.8$.

### D.4 ERRATIC ROBOT.

The 'Erratic robot' game takes place on a $4 \times 4$ grid map with 16 states and 5 actions, as shown in Fig. 9c. The robot is designed to minimize the number of steps required to collect goods while avoiding obstacle points. Specifically, its primary objective is to continuously take goods from designated destination points, earning a reward of $+5$ for each successful collection. The grid also includes four obstacle points, each incurring a penalty of $-2$ when encountered. The robot has five available actions: move left, move right, move up, move down, or remain stationary. Due to the instability, any movement of the robot has a probability $p \in [0, 1)$ of resulting in an additional unintended step. Actions that would move the robot outside the grid boundaries always fail. In this experiment, the probability of additional operation is set to $p = 0.2$. The results are presented in the Fig. 10.

As shown in the Fig. 10, the proposed *Act2cComm* scheme can achieve perfect communication with $R = 1/5$, albeit with a reduction in the average reward from the optimal value of $0.753$ to $0.45$. If we relax BER, the same rate can be achieved with a better reward. Similarly, for different reward targets, we can obtain different communication performances. This additional experiment also highlights the versatility of the proposed *Act2Comm* scheme across various environments.

## E ADDITIONAL ABLATION EXPERIMENTS

This section presents additional ablation experiments for our proposed *Act2Comm* framework, using the lucky wheel environment as the default experimental setting with an initial state $0$.

### E.1 ABLATION STUDY OVER DIFFERENT MECHANISMS

To showcase the efficiency of our approach, we present the loss values monitored throughout the iterative training process in Fig. 11. While the loss value occasionally fluctuates during the training updates—primarily due to the alternating update scheme between the decoder and encoder—overall, the model exhibits a rapid and consistent convergence.

Additionally, to further validate the performance of the Critic Network, we compare the estimated loss values generated by the Critic with the true loss during training. As shown in Fig. 11, the Critic Network maintains accurate loss estimation throughout the training phase.

We present a detailed comparative analysis of each mechanism within our method, as shown in Table 5, emphasizing the specific contributions of each component to the cumulative performance

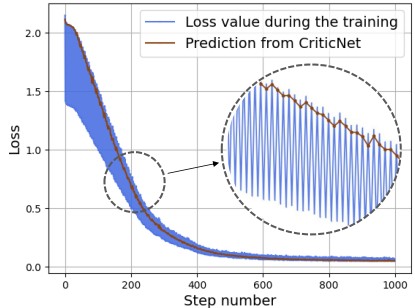

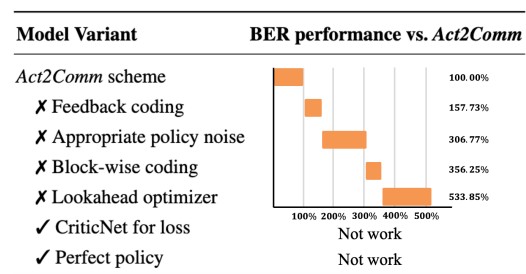

Figure 11: Loss during the training process, where the loss value decreases significantly with each decoder update, while showing a slight increase with each encoder update.

Table 5: Ablation study when sequentially removing one improvement after another. Note higher BER means worse performance for a given reward here with $R = 1/4$.

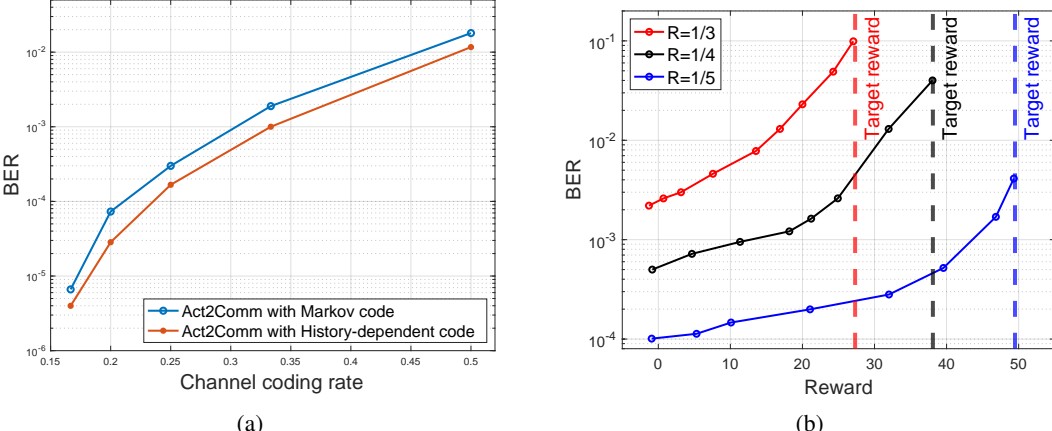

Figure 12: (a) Performance for different coding schemes. (b) Approaching a given policy $\bar{\bar{\pi}}$.

improvements. Notably, utilizing the Critic network to predict the loss directly, or eliminating policy noise during inner-step training can yield failure, underscoring the critical importance of appropriate policy noise and neighboring sampling mechanisms.

All these design elements collectively ensure that our proposed solution consistently delivers robust performance across a wide range of scenarios.

### E.2 ABLATION STUDY OVER THE FEEDBACK DESIGN

*Act2Comm* supports both history-dependent coding, which encodes using both message and feedback blocks, and Markov coding, which encodes using only the message block. We examine *Act2Comm* scheme with both history-dependent codes and Markov codes across various coding rates $R$ for a pure communication optimization.

The experimental results are depicted in Fig. 12a. It is evident that the BER performance improves significantly as the channel coding rate decreases. Specifically, with a coding rate $R = 1/6$, the BER reaches $10^{-6}$ level, indicating a significant enhancement in performance due to the increased number of channel uses. Comparing history-dependent codes with Markov codes reveals that incorporating feedback blocks can yield substantial performance improvements. Although, from a theoretical perspective, channel feedback does not enhance the capacity of a memoryless channel, it is beneficial for finite block-length coding. The observed benefits may be attributed to the attention mechanism applied to historical states and their corresponding actions. This mechanism effectively utilizes prior coding experience to simplify the subsequent coding process.

### E.3 Ablation study over the message length

Additional experiments with *Act2Comm* are conducted across different message lengths and coding rates, as presented in Table. 6. The results demonstrate that *Act2Comm* scheme consistently delivers competitive performance over different message lengths and coding rates. An interesting observation is that communication performance generally improves with longer block lengths; however, it eventually declines due to the increased learning complexity. Notably, for longer message lengths $k$, adapting a larger block size $\mu$ can potentially enhance performance while addressing the increased learning complexity.

| Coding rate | $k = 12$ | $k = 24$ | $k = 36$ |
|:---:|:---:|:---:|:---:|
| $R = 1/2$ | $1.16e^{-2}$ | $1.06e^{-2}$ | $9.15e^{-3}$ |
| $R = 1/3$ | $1.06e^{-3}$ | $1.02e^{-3}$ | $1.14e^{-3}$ |
| $R = 1/5$ | $2.84e^{-5}$ | $1.62e^{-5}$ | $2.06e^{-5}$ |

Table 6: BER for *Act2Comm* across different message length and coding rate, where $\mu = 3$.

### E.4 Ablation study over a specific policy

In the previous experiments, we used the optimal control policy as the target policy for training the encoder, as this is better for achieving a higher control reward. However, our approach is highly flexible and can be adapted to any control policy, allowing users to tailor the system to align with specific policies for their own tasks.

In the lucky wheel environment, now we consider a sub-optimal policy $\bar{\pi}$ as the target policy, given as:

$$\bar{\pi} = \begin{bmatrix} \pi(x=0|s=0) & \pi(x=1|s=0) \\ \pi(x=0|s=1) & \pi(x=1|s=1) \\ \pi(x=0|s=2) & \pi(x=1|s=2) \end{bmatrix} = \begin{bmatrix} 0.7 & 0.3 \\ 0.3 & 0.7 \\ 0.7 & 0.3 \end{bmatrix}. \tag{23}$$

We evaluate the *Act2Comm* scheme with different coding rates and consider a single message transmission with an accumulated reward. The experimental results in Fig. 12b display the curve representing the lower envelope of all possible BER-reward trade-off outcomes for each coding rate. This curve illustrates that the region above it is achievable by our *Act2Comm* scheme.

Notably, our method can achieve good communication performance, approximately $4 \times 10^{-1}$, $4 \times 10^{-2}$, and $3 \times 10^{-3}$ for $R = 1/3$, $R = 1/4$, and $R = 1/5$ respectively, with almost no loss in control performance. This is due to the stochastic nature of our target policy, as opposed to a deterministic one, allowing our method to adaptively learn a similar policy that maintains comparable rewards while being more favorable for channel coding. In summary, the experimental results demonstrate that our *Act2Comm* framework achieves satisfactory communication performance while maintaining the reward, as defined by a specific policy, at a certain level in scenarios such as those involving stochastic policies.

