# OpenReview forum: "Actions Speak Louder Than Words: Rate-Reward Trade-off in Markov Decision Processes"
_ICLR.cc/2025/Conference — ICLR 2025 Poster_

### Official Review · Reviewer_kFNj · 2024-10-18

**Soundness:** 3
**Presentation:** 2
**Contribution:** 3
**Rating:** 8
**Confidence:** 3

**Summary:**

This paper studies the fundamental tradeoff between the information content of an agent’s actions and the long-term performance of the agent in an MDP. This tradeoff is formalized as a convex optimization problem in the context of discrete state and action spaces. A transformer-based approach is then presented to approximate policies which solve a variant of the aforementioned problem, and results are presented in two small toy examples.

**Strengths:**

* The connection between information content and policy performance is of great theoretical and practical interest; this paper appears to provide a clean characterization of the tradeoff in finite state/action MDPs.
* The paper also proposes a deep learning-based scheme for generating policies which balance transmission rate and policy performance that—due to its use of neural networks—could (optimistically?) scale to larger problems.

**Weaknesses:**

* I was surprised not to see papers like “Least inferable policies for Markov decision processes” referenced in the related work discussion. There are other papers from some of these authors which, if memory serves, essentially try to find policies which balance entropy of the state distribution with expected future reward. I believe the main difference is that they were trying to maximize entropy of the state distribution, which amounts to making the capacity of your action-state channel very small. But I imagine it’s just a sign flip to get the other direction. I would appreciate some discussion here.
* Nitpick: line 131 should use \citep for the Blackwell reference. What does “indecomposable” mean?
* In the MDP preliminaries, don’t we need r() to be bounded in order to ensure that (1) is well-posed?
* nitpick: it would be helpful to define “unichain” for unfamiliar readers
* From only reading the main text, I would assume that Theorem 1 is surely known in the literature, and is a direct corollary of Shannon’s classic 1948 result, but adapted to the EAS channel. However, appendix A.1 presents a fairly involved proof. Please adjust the main text to explain what is new about this result from existing literature.
    * There should be some statement of proof for every theorem/lemma, or at least a clear statement that proofs can be found in a specific appendix. I did not see such a statement (apologies if I missed it).
* nitpick: it should be “ascent” not “decent” in line 306. Also some mention should be made of the inequality constraint here.
* I suggest that the authors have a bit more of a discussion around lemma 1 to explain how the concavity wrt V will be used later. Currently, lemma 1 interrupts the flow a little.
* nitpick: is the * in (12) multiplication? If so, I suggest using \cdot. Similar suggestion on line 418.
* The authors should clarify the discussion of non-differentiability of f_U in line 414 and below.
* “critic network with 3 linear layers” (line 431) -> does this imply that there are no nonlinearities? Perhaps: “fully-connected layers” is what is meant here?
* I am having a lot of difficulty following the logic of section 5 and would appreciate it if the authors could revise the presentation to more clearly articulate the key ideas, how they relate to the problem in Theorem 2, why each idea is necessary, etc.
    * For example, I do not understand what the role of the critic is, i.e. what its inputs and outputs are and how they relate to the problem at hand.
    * To be blunt, the first part of this paper (through section 4) is lovely, but section 5 is so confusing that it makes me seriously question my understanding of the problem from section 4. If this can be adequately addressed, I am very happy to raise my score. Specifically, I suggest that the authors:
       1. Provide a high-level overview of how the approach in Section 5 relates to and differs from the problem formulation in Theorem 2.
       2. Clearly explain the role of each component (e.g., the critic network) in the proposed approach, including its inputs, outputs, and how it contributes to solving the problem.
       3. Include a diagram or flowchart illustrating the relationships between different components of the proposed method.
       4. Consider reorganizing the section to first present the overall approach, then dive into the details of each component.

**Questions:**

* Reading the discussion around line 227, do we need for the code to be prefix-free for finite n?
* Do we really need Lemma 2?
* Why do the authors describe gradient descent for the problem in Theorem 2 when the proposed approach is actually solving a different problem? Theorem 2 is very nice to state, but I suggest the authors consider whether it is really necessary to discuss solving the problem in Thm. 2 in this level of detail if that is not what is actually done later in the paper.
    * This does beg the question of why not just use some gradient method to solve the problem in Thm. 2 instead of the proposed workflow outlined in section 5?
    * Put another way: why is the transformer-style approach in section 5 really necessary when the problem in Theorem 2 looks so elegant?
* Why are the regions in Figure 3b/c shaded? And, how are the target BER and reward handled? I did not pick up on that in the preceding discussion.
* The test environments are quite small. What is the main limitation in scaling the proposed approach? Specifically, please:
   1. Discuss the computational complexity of the proposed approach as the state and action spaces grow.
   2. Identify specific challenges you anticipate in applying the method to larger, more complex environments.
   3. Suggest potential modifications or extensions to the proposed approach that might address these scalability challenges.
   4. If possible, provide preliminary results or analysis on a slightly larger environment to demonstrate scalability potential.

---

> ### Author Response · Authors · 2024-11-21
> **Part 1 (Responses to main concerns)**
>
> We appreciate the reviewer’s valuable and constructive comments. To address the main concern about the relationship between Section 5 and Section 4, we would like to first clarify the distinction between **channel capacity** and **practical coding rate**, as this may not be familiar to researchers outside the fields of information and coding theory.
> As stated in Section 3, the channel capacity is defined as the maximum achievable rate for perfect transmission under **infinite code length** (i.e., $n\to \infty$). It characterizes the **performance limit** of the channel across all practical coding schemes.
> However, practical coding schemes have **finite code lengths**; and therefore, cannot achieve the capacity. The goal in practical code design is to achieve rates close to the capacity when  $n$  is large; or to minimize the error probability when $n$ is small since transmitting at rates close to capacity with an arbitrarily low probability of error is not possible when $n$ is small.
>
> In our paper, Section 4 is dedicated to characterizing the performance limit of communication via actions in MDPs from an information-theoretical perspective. We first derive the capacity of the action-state channel in Theorem 1 and then characterize the capacity-reward trade-off as a concave optimization problem in Theorem 2. In Section 5, we focus on designing a practical coding scheme (i.e., Act2Comm) for the finite block-length regime that balances the practical coding rate and the MDP reward.
>
> Now we can answer the reviewer's main question about **how the approach in Section 5 relates to and differs from the problem formulation in Theorem 2**. As mentioned above, the concave optimization problem in Theorem 2 characterizes the capacity-reward trade-off of the action-state channel. It provides an upper bound for practically achievable coding rates under a certain reward constraint. Solving the concave optimization yields the capacity and the optimal state-action distribution $\omega$, which can be translated to a stationary **control policy** for the MDP via the following formula:
>             $$
>                 \pi(a|s) = \frac{\omega(s,a)}{\sum_{a'} \omega(s,a')}.
>             $$
> Note that this policy $\pi$ cannot be directly used as the **coding policy** for communication. Instead we need to design a codebook specifying the codeword (i.e., a sequence of actions) to be transmitted for each possible message. This is similar to the relation between Shannon's capacity theorem and practical channel code design (e.g., Turbo, LDPC, polar codes) for more conventional channels.  Therefore, in Section 5, we propose Act2Comm to learn a coding policy that mimics the behavior of policy $\pi$ from the perspective of MDP control.
>
> In particular, Act2Comm generates a sequence of decision rules for each message, where each decision rule is a $|\mathcal{S}|$-dim vector specifying an action for each state. In the following time steps, the decision rules are used sequentially over time to determine an action based on the current state at each time step. Over the entire time horizon, the coding policy functions as a non-stationary control policy for the MDP. Using policy $\pi$ as the target policy, Act2Comm can effectively balance the rate-reward trade-off by learning a coding policy that aligns with the target policy $\pi$. Following the reviewers' suggestions, we have added a new section in the Appendix (C.1-C.6), which includes a detailed diagram illustrating each component of the proposed Act2Comm scheme. This section is also cited in the main body of the paper for clarity and reference.

---

> ### Author Response · Authors · 2024-11-21
> **Part 2 (Responses to weakness)**
>
> Next, we want to clarify **the novelty of Theorem 1 compared with Shannon's classic result**. Shannon’s classic channel capacity theorem characterizes the capacity of **memoryless channels**, where the channel output depends solely on the current channel input. In contrast, the action-state channel we study in this paper is a **finite-state channel (FSC) with memory**. Here, the channel output depends on both the channel input and the channel state, while the channel state itself depends on the historical inputs and states.
>
> In the related work section of the original manuscript, we included a brief review of studies on the capacity of FSCs. There are some known results that express the capacity of FSCs in **multi-letter forms**, but clear (single-letter) expressions for FSC capacity are generally unknown. Specifically, at the beginning of the proof of Theorem 1, we demonstrated that a multi-letter expression of the capacity of the action-state channel can be easily derived from existing results (see Equation (14) in Appendix B).
>
> However, the capacity expression given by Equation (14) involves the mutual information of two infinite sequences, rendering it uncomputable. In contrast, the capacity expression we present in Theorem 1 is simple and computable: $C=\max I(X;S^+|S)$. This is a surprising result, as it shows that the capacity of the action-state channel has a form similar to Shannon’s classic result. As we highlighted in the related work section, finding simple expressions for the capacity of general FSCs remains an open problem in information theory. In this paper, we utilize the special structure of the action-state channel to derive a **single-letter expression** for its capacity. We further show in Theorem 2 that the capacity of the action-state channel under reward constraint can be easily computed by solving a concave optimization. Therefore, we believe the capacity derivation provided in our paper is highly novel and surprising, and constitutes a valuable contribution to the literature on its own. We stated at the beginning of Section 4 that all proofs of this section are provided in Appendix B.
>
> In the following, we sequentially address the remaining comments regarding the weaknesses:
>
> * We thank the reviewer for bringing this interesting paper by Karabag at al. to our attention. After a thorough examination, we have determined that, while this work has some relevance, it is fundamentally different from ours. Their research also considers an MDP with an extra observer and uses information theoretic measures. However, the objective of their work is to find a control policy that can limit the ability of the observer to infer the transition probabilities of the MDP. They achieve this by formulating a problem to minimize the amount of information on the transition probabilities that can be gained from the state sequence.
>
>     In contrast, our work focuses on implicit communication between the controller and the observer. The message is encoded by the controller into the action sequence and decoded by the observer from the state sequence. Therefore, the observer's task is to infer the **action sequence** (rather than **transition probabilities**) from the state sequence. As we demonstrated in Section 4, the capacity of the action-state channel is equal to the maximum mutual information between action and state in the stationary distribution. It is not clear that there is a direct connection between the channel capacity and the amount of information about the transition probabilities derived from the state sequence. In the revised manuscript, we have cited this paper and clarified the difference between their work and ours.
>
> * We thank the reviewer for pointing out the citation format issue, which has been corrected in the revised manuscript. The term **indecomposable finite-state channel (FSC)** originates from information theory. In essence, an indecomposable FSC is an FSC for which the effect of the initial state dies away with time. For a precise and detailed definition, please refer to Robert Gallager's book ``Information Theory and Reliable Communication'', page 105.
>
> * You are right, the reward function $r(\cdot)$ should be bounded. We have modified the manuscript to clarify this issue.
>
> * We have added a sentence in the revised manuscript to explain the definition of ``unichain''.
>
> * Yes, “descent” should be “ascent” in line 306. We have corrected our statement in the revised manuscript.
>
> * Lemma 1 states a fundamental property of the rate-reward trade-off. We have added the following discussion in the revised manuscript to highlight the implication of Lemma 1:
>
>      *``Since the capacity is an upper bound for the rate of any practical coding scheme, Lemma 1 implies that the achievable region of rate-reward pairs forms a convex set.''*

---

> ### Author Response · Authors · 2024-11-21
> **Part 3 (Responses to questions)**
>
> In this section, we first address the remaining weaknesses and then answer the outstanding questions.
>
> * Yes, $*$ in (12) is multiplication. We have updated the revised manuscript to ensure that all multiplications are represented by the symbol `$\cdot$'.
>
> * The non-differentiability of $f_U(x|s)$ arises from its discrete representation. While the true values of $f_U(x|s)$ can be directly computed statistically, the gradient cannot propagate through to the encoder. We have clarified in the manuscript and detailed this process in Appendix C.
>
> * Yes, it should be ``fully-connected layers'' in line 431. We have corrected this.
>
> * To enhance the logical flow of Section 5, and following the reviewer’s suggestion, we have added a description to connect it with Theorem 2. We now begin the section with a high-level overview of the proposed approach, followed by detailed explanations of each component. Additionally, we have included a cited diagram in the Appendix, which points out the input, output, and functionality of each component.
>
> Finally, we answer the remaining unaddressed questions as follows:
>
> * A prefix-free code is a code in which no codeword is a prefix of another codeword. This property is desirable in **variable-length source coding**. However, our Act2Comm utilizes **fixed-length codes for channel coding**, meaning different messages are encoded into a fixed number of actions, and the message is decoded after receiving all channel outputs (i.e., states). Therefore, there is no need for it to be prefix-free.
>
> * As we discuss below Lemma 2, this result can be used to derive a closed-form expression of the gradient $\partial I /\partial \omega$, which is useful in solving the concave optimization problem in Theorem 2.
>
> * The shaded regions in Figure 3b/c represent achievable regions. As the proposed Act2Comm adopts a hyper-parameter $\lambda$ to balance the trade-off between control and communication, different $\lambda$ values result in distinct models, corresponding to different points in the figure. By selecting a target reward, various models with different $\lambda$ can achieve varying levels of communication performance. The shaded area in (c) illustrates the communication performance of a series of Act2Comm models that can achieve the given reward. Similarly, for a given communication requirement, the shaded area in (b), presents the achievable control performance of Act2Comm under the communication constraint. To enhance the clarity, we added these descriptions to the paper.

---

> ### Author Response · Authors · 2024-11-21
> **Part 4 (Responses to the last question)**
>
> The first part of the last question is addressed in Part 1. Here, we respond to the remaining aspects of the question.
> * According to the reviewer's suggestions, we first analyze the complexity of our approach with respect to increasing state and action spaces, as shown in Appendix C.6 (Tables 3-4). We can observe that an increase in the number of actions does not affect FLOPs or parameters, but a larger state space raises the encoder's computational complexity.
>
>     Note that the encoder must learn a codeword $\mathbf{U} \in \mathcal{X}^{|\mathcal{S}| \times \frac{k}{R}}$ for each message, where $k$ is the length of the message and $R$ is the coding rate. As the size of state space $\mathcal{S}$ and action space $\mathcal{X}$ grow, the code space expands exponentially, making it significantly more challenging to search for effective codes.  On the other hand, recall that the decoder in fact conducts a classification task over the sample space $\mathcal{X}^{|\mathcal{S}| \times \frac{k}{R}}$. Therefore, the expansion of $\mathcal{X}^{|\mathcal{S}| \times \frac{k}{R}}$ also increases the complexity of the decoding task.
>
>     Overall, the primary limitation in scaling the proposed approach is that expansions in the state and action spaces lead to an exponential growth of the code space, which, in turn, increases the difficulty of learning an effective policy. This insight highlights the challenges of applying the method to environments with very large state or action spaces.
>
> * Some potential solutions:
>
>     For complex environments with more actions, this challenge can potentially be addressed by incorporating more advanced architectures or coding mechanisms in the future. For example, a more sophisticated attention mechanism could be applied to the message block and feedback block, or a more advanced sequence-to-sequence learning approach could be utilized. Additionally, when dealing with larger action spaces, the encoder’s output could be regularized by introducing a regularization loss term. This term would increase the separation between different encoded action policy outputs in the belief map, further enhancing the model’s ability to handle complex action spaces.
>
>     For complex environments with more states, this challenge can potentially be addressed by incorporating more sophisticated architectures and extending training time with larger batch sizes. However, this yields more training cost and extra design of the model. As this is an initial pioneering effort, we leave the aforementioned optimization tailored for complex environments for future work.
>
>     Notably, increasing the batch size and training time allowed us to optimize the performance for the environment "Catch the Ball" with $p=0.2$, as demonstrated in the revised paper.
>
> * Due to the high training cost of larger models for environments with a large state space, it will be challenging to train a model for significantly more complex environments. However, to validate our method with more actions, we introduced a new environment, the ``Erratic Robot", with $5$ actions for additional experiments. Results demonstrate that our method remains effective, even in this more complex setting.

---

> > ### Comment · Reviewer_kFNj · 2024-11-25
> > **Thank you for the clear responses**
> >
> > I very much appreciate the clear explanations. I plan to increase my score.
> >
> > Only remaining nitpicks (which will not affect further scoring for me):
> > - I did not see any response to my question about prefix-free codebooks
> > - there is still an asterisk for multiplication in eq (12)

---

> ### Author Response · Authors · 2024-11-25
> **Response to the question about prefix-free codes**
>
> We greatly appreciate the reviewer’s prompt feedback and the improved score. We have previously addressed the question about prefix-free codebooks in Part 3 of our responses; you might have missed it due to the length of the reply. We will clarify this point again.
>
> A prefix-free code is one in which no codeword is a prefix of another codeword. This property is desirable in **variable-length source coding** because it ensures that each codeword can be uniquely and instantaneously decoded without ambiguity, as no codeword can be mistaken for the prefix of another. However, our Act2Comm utilizes **fixed-length codes for channel coding**, where different messages are encoded into a fixed number of actions (say, $n$ actions), and the message is decoded after receiving all the corresponding $n$ channel outputs (i.e., states). Since all codewords have the same length, it is impossible for one codeword to be mistaken for the prefix of another. Therefore, there is no need for the code to be prefix-free.
>
> Additionally, thank you for pointing out the asterisk issue. We have corrected it in equation (12) and proofread the manuscript again to ensure that all multiplications are represented by the symbol `$\cdot$’.

---

> > ### Comment · Reviewer_kFNj · 2024-11-25
> > **You are so right ;)**
> >
> > Sorry I missed the prefix free bit before. Makes sense. Thanks for the quick clarification!

---

### Official Review · Reviewer_ej1T · 2024-10-29

**Soundness:** 3
**Presentation:** 2
**Contribution:** 3
**Rating:** 6
**Confidence:** 3

**Summary:**

The proposed Act2Comm algorithm presents a multi-objective optimization for communication via the MDP state. This can be beneficial e.g. for multi agent scenarios in hostile environments where no dedicated communication infrastructure is available. In this case, the policy acts both as a message encoder as well as a controller. This requires integrating communication (such that a certain code rate and error probability is achieved) and control (such that a minimum return is achieved). Act2Com presents a transformer-based approach to this challenge.

**Strengths:**

S1: The idea appears to be novel and has not yet been addressed before.

S2: The paper is mostly written well.

**Weaknesses:**

W1: The paper requires further clarification and explanation of several critical aspects as lined out in the questions.

W2: The setting relies on finite state and action spaces and does not yet address multi-agent scenarios.

W3: No code base provided.

W4: The structure is not ideal in that Theorem 1 is introduced in Section 4, but the proof of Theorem 1 requires results from Section 5.

Further (minor) weaknesses:
* Figure 1 is pretty crowded and does not look nice
* Typo l. 131 „Blackwell Blackwell“
* Typo l. 229 „use“s

**Questions:**

Q1: Why is C(\tau) necessary in the Transceiver, given Theorem 1 states no historical data is required for the encoding?

Q2: Theory (e.g. Theorem 2) is limited to the undiscounted case, which can cause issues in non terminating MDPs as the return values are likely to diverge to plus or minus infinity. At the same time, the communication does not account for terminal states in the MDP, thus I assume the work considers non terminating MDPs. This needs to be clarified or addressed.

Q3: l348 what is L? Generally, I think the introduction of the decision rule U is relatively confusing. From what I understood, U is a rule that given the current channel state provides the corresponding action that results in the required next state the decoder needs to observe to transmit the message according to the code of the EAS channel. Is that correct? If so, I think an introductory sentence along these lines would benefit the paragraph.

Q4: l382 Are the results limited to scalar states and actions? Otherwise I do not understand the dimensionality of c_t^{(\tau)}.

Q5: l398: I am not really sure how the quantization works. Shouldn't it be something like U = Q(Z) = |x| \cdot Sigmoid(Z) for all x in X?

Q6: L452 The paragraph is unclear. Is the estimated gradient the one of eq. 10? If so, why does the critic approximate precisely this gradient and not something else?

Q7: How does your work differ from [1]? The paper should discuss this in my opinion.

[1] Communicating via Markov Decision Processes, Sokota et al. ICML 2022

---

> ### Author Response · Authors · 2024-11-21
> **Part 1 (Responses to weaknesses and the first two questions)**
>
> We thank the reviewer for the careful review and insightful questions, which have helped refine the manuscript.
>
> * **W1**. We have revised the paper to further clarify and explain the aspects mentioned in the reviewer's questions. See our responses to the questions for details.
>
> * **W2**. It is correct that this paper focuses on MDPs with finite state and action spaces. The primary reason for this restriction is due to communication considerations. The main novelty of this work is to treat the MDP environment as a communication channel, enabling us to encode messages into the action sequence and decode them from the state sequence. An MDP with finite state and action spaces can be viewed as a finite-state channel (FSC), a complex setting that poses significant challenges for both capacity characterization and the design of practical coding algorithms. Extending this approach to continuous MDP environments would result in memory channels with continuous states (affected by channel inputs), which presents even greater challenges. Since this is the first work on communication via actions in MDPs, we believe that starting with a setting of finite state and action spaces is a reasonable and practical formulation. Moreover, there are numerous practical applications that involve finite state and action spaces, so we believe this restriction does not diminish the significance of our work.
>
>   The proposed concept of communication via actions can be applied to multi-agent scenarios, particularly when dedicated channels are not available. In such cases, we need to expand our current framework to account for how the receiver utilizes the messages and how this impacts the overall system. This presents an intriguing and challenging direction for future research.
>
> * **W3**. We will submit our code along with detailed documentation in the supplementary material after posting this response.
>
> * **W4**. The proof of Theorem 1 relies on the equivalence between the **action-state channel** and the **extended action-state (EAS) channel**. Specifically, we demonstrate that the original action-state channel can be equivalently converted to the EAS channel. We initially presented this equivalence in Section 5 for two main reasons: first, the proof of Theorem 1 is provided in the Appendix, and understanding this equivalence is not necessary for comprehending Theorem 1; second, this equivalence is crucial for the design of Act2Comm. To address the reviewer's concern and improve readability, we now state this equivalence at the beginning of the proof of Theorem 1 in the Appendix.
>
> * **Minor Weakness**: We have re-plotted Figure 1 and corrected the typos.
>
> We next answer the reviewer's questions.
>
> * **Q1**. Yes, we established in Theorem 1 that no historical data is required for achieving the channel capacity. Please note that the channel capacity is defined as the maximum achievable rate for perfect transmission under infinite code length (i.e., $n\to \infty$, see Section 3). It serves as an upper bound for all practical coding schemes. However, practical coding schemes have finite code lengths, and therefore, they generally cannot achieve the capacity but can achieve rates close to the capacity when  $n$  is large. While historical data is not necessary to achieve the capacity in the infinite block length regime, it can be beneficial for achieving better coding performance in the finite block length regime.
>
>     For example, feedback does not increase the capacity of a memoryless channel, and hence, it can be ignored from a capacity point of view [1-2]. However, as demonstrated in [3], it can simplify the coding mechanism and improve the error probability. This is exactly the motivation why we have introduced $C(\tau)$.
>
>     [1]. Shannon, Claude. ``The zero error capacity of a noisy channel." IRE Transactions on Information Theory 2.3 (1956): 8-19.
>
>     [2]. Dobrushin, Roland L'vovich. ``Information transmission in a channel with feedback." Theory of Probability \& Its Applications 3.4 (1958): 367-383.
>
>     [3] Schalkwijk, J. ``A coding scheme for additive noise channels with feedback--II: Band-limited signals." IEEE Transactions on Information Theory 12.2 (1966): 183-189.
>
> * **Q2**. Yes, as defined in Section 3, this paper considers non-terminating MDPs with the long-term average reward criterion. Following the convention, we assume that the MDP is unichain and that the reward function is bounded. In this setting, although the total reward over an infinite time horizon would diverge, the long-term average reward (defined in equation (1)) is guaranteed to be bounded. We have added a sentence in the revised manuscript to emphasize this setting.

---

> ### Author Response · Authors · 2024-11-21
> **Part 2 (Responses to Q3 to Q6)**
>
> In this part, we answer the reviewer's questions 3-6.
>
> * **Q3**. The use of $L$ here was a typo, which has been corrected into $|\mathcal{S}|$. The dimension of the decision rule should correspond to the number of states, meaning that each generated decision rule specifies the decision made for the corresponding future states.
>
>     The reviewer’s understanding is correct. In block coding, a sequence of decision rules is generated for the next several time steps during each coding round. They will be sequentially used to determine an action once a new state is revealed. Following the reviewer’s suggestions, we have revised the following sentences to enhance clarity.
>
>   *``Specifically, a decision rule specifies an action for each state; hence it can be used to determine an action for any given state.''*
>
> * **Q4**: The results are not limited to scalar states and actions; the dimensionality shown here serves as an example based on the specific environment discussed later. For vector states and actions, the feedback block can be constructed by encoding each component separately and subsequently concatenating them. Alternatively, vector states or actions with finite values (as in our FSC scenario) can be equivalently indexed as scalar values for simplicity.
>
>     As supplementary evidence, the feedback block is capable of embedding all historical states (as vectors) and policy matrices from our experiments. However, for simplicity and effectiveness, the paper utilizes only the information from the current state, the selected action, and the subsequent state. This results in the feedback vector being defined as $\mathbf{c_t^{(\tau)}} \triangleq [s_t^{(\tau)}, x_t^{(\tau)}, s_{t+1}^{(\tau)}] \in \mathbb{R}^{1 \times 3}$ for each time step.
>
>     To enhance clarity and comprehension, we have included pseudocode for the algorithm in Appendix C, including detailed coding schemes, codeword definitions, and the feedback block update strategy.
>
> * **Q5**: The quantization function, $\mathcal{Q}: \mathbb{R}^{|\mathcal{S}|\times \frac{k}{R}}\rightarrow \mathcal{X}^{|\mathcal{S}|\times \frac{k}{R}}$, is designed to convert continuous coding results into feasible decision rules. Note that a decision rule is a vector with the $i$-th element being an action for state $i$.
>
>     We note that the codeword $\mathbf{U}=\mathcal{Q}(|\mathcal{X}| \cdot \text{Sigmoid}(\mathbf{Z}))$ must belong to the finite input alphabet, where $|\mathcal{X}|$ represents the total number of actions. Specifically, the continuous coding result, $|\mathcal{X}| \cdot \text{Sigmoid}(\mathbf{Z}) \in (0,|\mathcal{X}|)$, must be mapped into the discrete input alphabet of the channel. To achieve this, we use a quantizer that rounds down each element of the coding result to the nearest action index. For example, if $|\mathcal{X}| \cdot \text{Sigmoid}(\mathbf{Z}) =[1.5,0.8,2.1,3.2,4.8]$, it will be rounded to $[1,0,2,3,4]$ as the channel input.
>
>   We have added a paragraph in Appendix C of the revised manuscript to further explain how the quantizer works.
>
> * **Q6**: This paragraph illustrates how the proposed Act2Comm scheme is optimized. This iterative training strategy is designed to address the non-differentiability of the FSC. Gradient estimation involves approximating the gradient required for encoder updates during backpropagation. Since the gradient cannot flow directly back to the encoder, a critic network is employed to accurately predict the decoder's output, enabling the gradient to flow back through the critic network and facilitating effective encoder optimization.
>
>     During encoder optimization, the critic network is first trained to approximate the unknown environment, which comprises the quantizer, the FSC channel, and the frozen decoder. Once trained, the critic network then assists in updating the encoder. Subsequently, the encoder is frozen, and the decoder is updated to determine the optimal decoding strategy for the fixed encoder. With the updated decoder, the environment changes, necessitating the updating of a new critic network to support the next round of encoder updates. This iterative process is repeated until all components converge. The effectiveness of this approach is visualized in a figure provided in the appendix. We want to mention that, in principle, a critic network can be designed to directly predict the loss values, as commonly utilized in RL algorithms. However, based on our ablation study (Appendix, Table 5), a critic network designed to predict the logits from the decoder proves to be the most effective.
>
>   For further clarification, an additional section with a diagram and pseudocode which illustrate the gradient flow is included in Appendix C of the revised manuscript.

---

> ### Author Response · Authors · 2024-11-21
> **Part 3 (Response to Q7)**
>
> We answer the last question in this part.
>
> * **Q7**: Many thanks to the reviewer for bringing this interesting paper to our attention. After carefully reading it, we found this work to be highly related to ours. We have included the following discussion in the revised manuscript to highlight the similarities and differences between their work and ours:
>
>     *``Sokota et al. (2022) explored a similar concept of communication via MDPs. In their study, the receiver can observe the entire trajectory, including both the action and state sequences. This enables the controller to encode (compress) messages into the action sequence, and the receiver can subsequently decode the messages from the trajectory. Essentially, this is a randomized source coding problem. However, in most practical scenarios, while the MDP state is a physical signal observable by the receiver, the controller’s actions are usually not directly observable to other agents. Therefore, in our work, we assume the receiver can only observe the state sequence. This shifts the problem from source coding to channel coding.''*
>
>      More specifically, in the Sokota et al. paper, a message $M$ is mapped into an action sequence $(a_1,a_2,\cdots,a_n)$, resulting in a state sequence $(s_1,s_2,\cdots,s_n)$; the receiver then infers the message from the trajectory $(a_1,s_1,a_2,s_2,\cdots,a_n,s_n)$. To optimize this inference, Sokota et al. employ the minimum entropy coupling technique to maximize the mutual information between the message and the trajectory, enhancing message inference from the trajectory.
>
>    In contrast, our paper addresses a channel coding problem where the receiver needs to infer the action sequence from the state sequence. Ideally, the encoder provides a one-to-one mapping from the message set to the action sequences, ensuring that the transmitted message can be accurately determined once the action sequence is inferred from the state sequence. As demonstrated in Theorem 1 (Section 4), the channel capacity is given by the mutual information between the action and state in the stationary distribution. The main challenge in our problem is to select a unique action sequence for each message such that the mutual information between action and state is maximized.
>
>     Moreover, we establish theoretical results for this class of problems and propose a practical coding scheme. We also evaluate our approach in several environments that are more complex and more practical than those previously considered.

---

> > ### Comment · Reviewer_ej1T · 2024-11-25
> > **Reply to authors' response**
> >
> > I want to thank the authors for their detailed response. While most of my concerns have been addressed, a few points need further clarification:
> >
> > ### Q3:
> >
> > The reformulated paragraph is clearer than before, but still does not connect the concept of the decision rule and the control policy extremely well.
> >
> > ### Q4:
> >
> > From the text, it is not clear that the dimensionality refers to some 1-dimensional example. This either needs to be clarified or (preferably)  $dim(s) = n_s$ etc. needs to be introduced to provide general dimensionality expressions.
> >
> > ### Q6:
> >
> > I acknowledge the effort to make this part clearer, however, the connection to the theoretical results of Section 4 remains vague. I think the paper would largely benefit from a clear description which algorithmic building block approximates which part of the theoretical results.
> >
> >
> > Even though there are still problems with the paper that should be addressed, I increase my score as the revised version of the paper represents a substantial improvement.

---

> ### Author Response · Authors · 2024-11-26
> **Part 1 (Second-round Responses to Q3 and Q4)**
>
> We greatly appreciate the reviewer for the follow-up comments and the improved score. Your feedback is invaluable for clarifying key concepts and enhancing the presentation of our paper. Therefore, we have revised the manuscript accordingly to address your concerns.
>
> * **Q3**. We have revised the corresponding paragraph in the updated manuscript to better clarify the relationship between the decision rule and the control policy:
>
> >``Let $\mathcal{U} \triangleq \mathcal{X}^{|\mathcal{S}|}$, where each ${u} \in \mathcal{U}$ is referred to as a decision rule because the $i$-th element of $u$ can be viewed as an action prescribed for state $i$. A deterministic control policy can be defined as a sequence $\{u_t:t\ge 1\}$, where $u_t\in \mathcal{U}$ is the decision rule at time $t$. To facilitate block coding, we first convert the \textit{action-state channel} into an extended action-state (EAS) channel (Fig. 6 in Appendix) using Shannon's method (Shannon, 1958). In particular, $\cdots \cdots$ This approach allows us to map a data block to a sequence of decision rules without knowing future states. The actions for subsequent time steps can then be determined using these decision rules when the states are revealed.'' (Section 5, page 7)
>
> * **Q4**. In this paper, we focus on MDPs with finite state and action spaces, without restricting to MDPs with scalar states and actions. In Experiment 2, for instance, the state comprises two components: the position of the board, denoted by $a_t$, and the position of the ball, denoted by $b_t$. A natural way to represent the MDP state is as a vector $(a_t,b_t)$.
>     Alternatively, since the state and action spaces are both finite, we can map the vector states to scalar states by indexing them. For example, in Experiment 2,  $a_t$ can take values in $ \{0,1,2\}$ and $b_t$ in $\{0,1,2,\cdots,8\}$. Hence the vector $(a_t,b_t)$ has 27 possible combinations. We index these 27 vectors by integers $0,1,2,\cdots, 26$ and use these indices as the MDP states. For ease of presentation, our current Act2Comm adopts this latter approach.
>     We mentioned this in the first paragraph of Section 5: `The sets of states and actions are indexed as $\mathcal{S}=\{0,1,\ldots, |\mathcal{S}|-1\}$ and $\mathcal{X}=\{0,1,\ldots, |\mathcal{X}|-1\}$'. Consequently, states and actions in Act2Comm are represented by scalars, even when they originate from vector states and actions. This is why we wrote ``$\mathbf{c_t^{(\tau)}} \triangleq [s_t^{(\tau)}, x_t^{(\tau)}, s_{t+1}^{(\tau)}] \in \mathbb{R}^{1 \times 3}$'' in the original manuscript.
>
>     Inspired by the reviewer’s comment, we modified Act2Comm to adopt the vector state representation. This change is minor and can be implemented by slightly adjusting the feedback structure to the encoder and the input structure to the decoder. In this case, the feedback vector for the $\tau$-th round is given by $\mathbf{c_t^{(\tau)}}\triangleq[s_t^{(\tau)},x_t^{(\tau)},s_{t+1}^{(\tau)}] \in \mathbb{R}^{1 \times (2n_s+n_x)}$, where $n_s$ is the state dimensionality and $n_x$ is the action dimensionality.
>     We then tested this variant in Experiment 2. The results showed that using the vector state representation, the communication loss converged slightly faster than with the scalar state representation. However, the final policies learned by both methods were nearly identical. An intuitive explanation for this result is that the vector state representation captures more detailed features of the environment, which may potentially facilitate learning. We have included a discussion of these findings in Appendix C.5.

---

> ### Author Response · Authors · 2024-11-26
> **part 2 (Second-round Response to Q6)**
>
> This part provides our response to **Q6**:
>
> To clarify the relationship between the theoretical results in Section 4 and the algorithm in Section 5, we must first distinguish between **channel capacity** and **practical coding rate**, which may not be familiar to researchers outside the fields of information and coding theory.
> As stated in Section 3, the channel capacity is defined as the maximum achievable rate for perfect transmission under **infinite code length** (i.e., $n\to \infty$). It characterizes the performance limit of the channel across all practical coding schemes.
> However, practical coding schemes have **finite code lengths**; and therefore, cannot achieve the capacity. The goal in practical code design is to achieve rates close to the capacity when  $n$  is large, or to minimize the error probability when $n$ is small since transmitting at rates close to capacity with an arbitrarily low probability of error is not possible when $n$ is small.
>
> In our paper, Section 4 is dedicated to characterizing the **performance limit** of communication via actions in MDPs from an information-theoretical perspective. We first derive the capacity of the action-state channel in Theorem 1 and then characterize the capacity-reward trade-off as a concave optimization problem in Theorem 2. Additionally, we show that this concave optimization can be efficiently solved using gradient ascent methods, thanks to the closed-form expression of gradient implied by Lemma 2. In Section 5, we shift our focus to **designing a practical coding scheme**, Act2Comm, for the finite code-length regime. This scheme aims to balance the practical coding rate and the MDP reward.
>
> It is important to note that solving the concave optimization only yields the capacity and the optimal state-action distribution $\omega$, but does not provide a practical coding policy for communication. This situation is similar to the relationship between Shannon’s capacity theorem and practical channel code designs (e.g., Turbo, LDPC, polar codes) for more conventional channels.
> Therefore, the algorithm in Section 5 is not developed to solve the optimization problem in Theorem 2, nor does it introduce algorithmic components to approximate aspects of the theoretical results. Instead, Section 5 focuses on designing a practical coding scheme that can effectively balance the coding rate and MDP reward within a **finite block-length** regime.
>
> We have revised the first paragraphs of Section 4 and Section 5 to highlight the tasks of these two sections: in Section 4,
>
> >``In this section, we analyze the trade-off between the capacity of the *action-state channel* and the MDP reward. While the results may not offer direct guidance for practical coding---since the capacity is typically achievable only in the infinite-horizon regime (i.e., when the code length $n\to \infty$)---they delineate the fundamental performance limits of communication via actions in MDPs, thus holding substantial theoretical importance.'' (Section 4, page 5)
>
> In Section 5,
> >``This section presents *Act2Comm*, a learning-based practical coding scheme that balances both control and communication objectives. This framework assumes a pre-determined control policy $\pi$ that satisfies the reward constraint $G_\pi \ge V$, referred to as the target policy. Such a target policy can be easily derived using traditional MDP or RL algorithms. Alternatively, solving the problem in Theorem 2 yields a policy $\pi$ that precisely satisfies $G_\pi=V$. *Act2Comm* aims to learn a coding policy that: (1) closely mimics the stochastic behavior of the target policy;
> (2) minimizes the probability of decoding errors for a given coding rate and a finite code length. That is, *Act2Comm* takes a policy achieving the desired reward, and embeds messages into it with the desired reliability.'' (Section 5, page 7)
>
> We have also highlighted the relationship of these two parts in the Introduction:
> >``We also show that the capacity-reward trade-off can be characterized by a convex optimization, which can be solved numerically. While the capacity-reward trade-off provides an upper bound on the practically achievable rate under certain reward constraints, solving it does not yield a practical coding scheme. We then propose a practical framework for the integrated control and communication task in the finite block-length regime.'' (Section 1, page 2)

---

### Official Review · Reviewer_m4h4 · 2024-10-31

**Soundness:** 3
**Presentation:** 3
**Contribution:** 2
**Rating:** 5
**Confidence:** 4

**Summary:**

This paper considers the problem of communication in decision-making problems, they consider communicating through actions, where the message is embedded into the actions of the agents for interacting with environments. The authors model the MDP environment as a finite-state channel (FSC), with actions as the input and the states of the receiver as the output. They propose a way to balance the long-term reward and the communication compacity and show that they could achieve both optimal policies for return and minimize the communication cost at the same time.

**Strengths:**

1. This paper is well-written and well-organized.
2. The proposed Act2Comm effectively balances the dual objectives of control for total return and communication cost.

**Weaknesses:**

1. Act2Comm assumes a pre-determined control policy $\pi$, which means it also needs to optimize the gap between this target policy and the Act2Comm learned policies. Could add a return gap part to further stabilize this procedure. Refer to this paper here 'Chen, J., Lan, T., & Joe-Wong, C. (2024). RGMComm: Return Gap Minimization via Discrete Communications in Multi-Agent Reinforcement Learning. Proceedings of the AAAI Conference on Artificial Intelligence, 38(16), 17327-17336. https://doi.org/10.1609/aaai.v38i16.29680'.
2. I suggest the authors add a pseudo code in the appendix to further assist the readers to know the whole training process. Right now the order of training the control policies and the communication messages is unclear.
3. The testing environments are too simple for testing MARL-Comm methods, which only consist of 3 states or 27 states, and 2 or 3 discrete actions. The methods should be tested in continuous control environments.
4. There are no baselines for comparing the Act2Comm algorithms for the empirical evaluation. This may affect the theoretical results' soundness.
5. I did not find the submitted codebase in this paper's submission files. The reproducibility might be a problem.

**Questions:**

1. The whole process is very similar to some existing works for multi-agent communications, such as: "Lowe, R., Wu, Y. I., Tamar, A., Harb, J., Pieter Abbeel, O., & Mordatch, I. (2017). Multi-agent actor-critic for mixed cooperative-competitive environments. Advances in neural information processing systems, 30", these works are also training the messages along with actions as well. Could the author further discuss a bit more about how it differs from the existing implicit communication methods which are training the messages with actions?
2. This paper models the MDP, if I understand it correctly, it should be a multi-agent MDP, then is this MDP fully observable or partially observable? Then this MDP should be modeled as the multi-agent version, and should define the number of agents and also the joint state, observation, and action space. Could the authors further clarify this part?
3. How the communication messages like? Are these discrete or continuous?

---

> ### Author Response · Authors · 2024-11-21
> **Part 1 (Clarification and responses to Weaknesses)**
>
> We thank the reviewer for the valuable comments and for recommending two interesting papers. First and foremost, we would like to respectfully clarify that the problem studied in our paper is not a multi-agent MDP, nor is the proposed Act2Comm a MARL algorithm. Although our setting involves two agents (the controller and the receiver), the MDP is influenced solely by the controller’s actions. The receiver acts as a **passive observer**, merely observing the MDP states to decode the message without taking any actions to control the MDP. Additionally, the use of the received message at the receiver is not specified and is beyond the scope of this paper. Our focus is on the concept of **communication through actions**, where the controller encodes the message into the action sequence and the receiver decodes it from the state sequence. This approach allows us to achieve a certain level of rewards from the MDP control while simultaneously enabling message communication from the controller to the receiver. The proposed Act2Comm is a framework to balance the dual objectives of control (aimed at maximizing rewards) and communication (aimed at achieving higher communication rate and reliability).
>
> Next, we will address the reviewer’s comments in detail.
>
> * **Weakness 1**. We thank the reviewer for the suggestion; however, we would like to clarify that our approach inherently incorporates a return gap component similar to the one discussed in Chen et al. (2024). In their work, they derive an upper bound for the return gap and minimize it to learn a policy that closely approximates the ideal policy.
>     In our paper, we train Act2Comm by minimizing a loss function $\mathcal{L}=$ communication loss + $\lambda$ control loss, where ``control loss'' quantifies the distance between the learned policy and the target policy. Leveraging perturbation analysis techniques (see, e.g., X. R. Cao, Stochastic Learning and Optimization: A Sensitivity-based Approach, Springer), we can establish a bound on the return gap for two policies. This bound approaches zero as the distance between the two policies decreases. Therefore, by minimizing the control loss term, we can effectively minimize the return gap between the learned policy and the target policy. The proposed Act2Comm method balances the dual objectives of control and communication by minimizing a weighted sum of communication loss and control loss.
>
> * **Weakness 2**. For the proposed Act2Comm scheme, we assume a given target policy (which can be the optimal control policy or the optimal tradeoff policy obtained by solving the concave optimization in Theorem 2). Then we train the Act2Comm to learn a coding policy that balances control rewards and communication performance using a weighted loss function, without enforcing a training order.
> In response to the suggestion, we have added the pseudocode (Algorithms 1 and 2) for the training and inference processes in Appendix C of the revised manuscript. Additionally, we have included a new section in the Appendix with diagrams to further illustrate the training process.
>
> * **Weaknesses 3\&4**. As clarified above, the proposed Act2Comm is not a MARL-Comm method. Instead, this paper explores the novel concept of communication through actions in an MDP while simultaneously maximizing the average rewards. From an information theory perspective, we treat the MDP environment as a finite-state channel (FSC). We derive the capacity of this FSC and characterize the trade-off between communication capacity and control reward. We focus on MDPs with finite state and action spaces because a continuous MDP environment, which corresponds to a memory channel with continuous states and inputs, presents significant challenges in terms of both capacity characterization and the design of practical coding algorithms.
>     Since our problem formulation is novel, there are no benchmarks for comparison, to the best of our knowledge. Nevertheless, our theoretical results regarding the channel capacity and the capacity-reward trade-off are general, and have rigorous theoretical proofs. These can be of independent interest to better understand the fundamental limits of similar channels.  Additionally, we have conducted extra experiments for an MDP with more states and actions to verify the effectiveness of Act2Comm. The experimental results are provided in Appendix D.4 (Erratic robot).
>
> * **Weakness 5**: We will submit our code in the supplementary material after posting this response.

---

> ### Author Response · Authors · 2024-11-21
> **Part 2 (Responses to questions)**
>
> We answer the reviewer's questions in this part.
>
> * **Question 1**. In most existing studies on multi-agent communications, a **dedicated channel** is assumed for communication between agents. When the communication bandwidth is insufficient to transmit the original observations, each agent must learn what to transmit at each time. For example, in the work of Chen et al. (2024) (the paper mentioned in the reviewer's comment), each agent learns a message generation function. Once the message is generated by this function, it is assumed to be reliably transmitted to other agents via the dedicated channel. The problem of determining what to transmit is essentially a **source coding** problem. The communication setting in Lowe et al. (2017) is even simpler; they assume sufficient communication bandwidth such that agents can directly communicate their observations to each other, allowing each agent to use the observations of others to infer their actions.
>
>     In contrast, our paper explores communication without the use of dedicated channels. Specifically, we demonstrate that an MDP environment itself can be used as a communication channel. In this approach, messages are encoded into action sequences and decoded from state sequences. The proposed Act2Comm method learns a joint control and **channel coding** policy, which selects actions to both encode messages into the action sequence and maximize the MDP reward simultaneously. It is important to note that in our problem, the message action and the control action are the same, so encoding a message directly affects the agent’s own actions. In contrast, in the previously mentioned examples of multi-agent communication, the message action and control action are distinct, and the transmitted message primarily affects the actions of other agents (the receivers). We have cited the aforementioned papers in the related work section and have clearly distinguished our work from those addressing multi-agent communication via dedicated channels.
>
>     Our framework can be applied in multi-agent systems to enable communication between agents without dedicated channels, potentially improving multi-agent control performance. This represents an interesting yet challenging direction for future research.
>
> * **Question 2**. As clarified in the first paragraph of part 1, the problem studied in this paper is not a multi-agent MDP. Instead, the system consists of a single-agent MDP with an extra passive observer (i.e., the receiver) that does not influence the MDP. The state of the MDP is fully observable to both the MDP controller and the receiver.
>
> * **Question 3**. The messages in our system are discrete (i.e., bits). In modern digital communication systems, source coding and channel coding are two key components. Source coding handles data compression; when messages are continuous, the source encoder quantizes them into discrete messages. Channel coding then maps these discrete messages into a sequence of channel inputs in order to protect them against channel noise. In this paper, we frame the problem of communication via actions as a joint control and channel coding problem. Therefore, we assume that the message set is finite, with each message to be transmitted randomly sampled from this set.

---

### Official Review · Reviewer_dvdt · 2024-11-04

**Soundness:** 2
**Presentation:** 2
**Contribution:** 2
**Rating:** 8
**Confidence:** 2

**Summary:**

The paper presents an analysis and method to allow RL agents to transmit messages through control actions as sequences of visited states, trading off average reward for information rate of the transmitted message. They provide theoretical results on the maximum information theoretic capacity such transmissions have, and propose a model and algorithm to do this efficiently.

**Strengths:**

The main strengths of the paper are:
- The problem and issues addressed are well stated and clear.
- The theoretical results presented are interesting.
- To the best of my knowledge, encoding messages in state-action sequences of RL agents is a novel idea.

**Weaknesses:**

The main weaknesses are:
- I am not convinced of the significance of the work. Although the problem is relatively interesting, the theoretical results seem to fall a bit short, and the experiments are not very illustrative either (see questions below).
- The notation is a bit cluttered and makes some technical statements hard to follow. I understand that authors make use simultaneously of RL notation and FSC notation, but as is, it is confusing. Eg: $x$ is used for actions, but in line 170 there is $a_t$ which seems to be used for actions too. $\Epsilon$ and $\pi$ seem to be used for policies, sometimes in the same statement. In line 449 there is a $\mathcal{L}_{cn}$ which is not defined. In Appendix 1, the proof for the main theorem in Section 4 is presented, but immediately after the authors state that this section makes use of results in section 5 for the proof. $s^+$, $s_{t+1}$ and $s^{t+1}$ seem to be used for the same variable.
- It is not obvious to distinguish whether the experiment environments are a common benchmark in other information theoretic RL work, or they have been specifically selected for this particular application. I understand that the method is novel, and therefore there may not be other benchmarks to compare to, but some justification for the particular choice of the experimental tasks would be beneficial.

**Questions:**

- What would be a general motivating example for wanting to trade off rewards for information rates in the state sequences?
- The theoretical results in page 6 revolve around concavity with respect to the joint stationary distribution. However, we usually cannot directly solve this, since the only control variable is the policy. The stationary state distribution in an mdp is an infinite product of transition matrices, which depend linearly on the policy. How do these results translate to policies? If they don’t, are they really useful?
- Section 5 needs some intuition. After the theoretical results, the particular model architecture choice is not obvious.
- In line. 452, it is stated that at each step of the algorithm, a critic needs to be trained to estimate the gradient. How costly is this?

---

> ### Author Response · Authors · 2024-11-21
> **Part 1 (Summary of modifications)**
>
> We thank the reviewer for their valuable comments. We have diligently revised and improved the manuscript, and believe that it has been significantly enhanced. Before providing detailed descriptions, we offer a summary of the major modifications made in the current version.
> * We have further simplified and unified the notation and included a notation table in the Appendix for clarity.
> * We have added an additional experiment (``Erratic Robot'') to validate the versatility of the proposed Act2Comm method.
> * Following the concerns from the reviewer, we have added a new section in the Appendix that includes detailed implementation descriptions, algorithms, complexity analysis, and training illustrations.
> * We provide the code as supplementary material. A project website (currently anonymous) will be released, which additionally provides the demo, training logs, and supplementary materials to facilitate reproducibility and future work.
>
> Following this, we will present detailed point-by-point responses to each weakness and question.

---

> ### Author Response · Authors · 2024-11-21
> **Part 2 (Responses to Weaknesses)**
>
> This part provides point-to-point response to the weakness:
> * **Weakness 1.** This paper explores a novel concept of communication via actions in MDPs, demonstrating that messages can be transmitted from the MDP controller to a receiver without the need for a dedicated communication channel. That is, the MDP environment itself can function as a communication channel (referred to as the action-state channel), enabling messages to be encoded into the action sequence and decoded from the state sequence. To this end, we formulate a joint control and channel coding problem to investigate the trade-off between the control and communication objectives.
>
>     From an information-theoretical perspective, we derive a single-letter expression for the capacity of the action-state channel in Theorem 1. We believe this is a significant result that can be of independent interest. We would like to note that channel capacity can be obtained only for a very limited set of channels. This serves as a theoretical upper bound on the amount of information that can be reliably transmitted over this channel. In Theorem 2, we characterize the capacity-reward trade-off as a concave optimization problem, which can be efficiently solved using the gradient ascent method thanks to the implication of Lemma 2. Together, these theoretical results provide a comprehensive characterization of the fundamental trade-off between the communication rate through actions and the average reward in MDPs. We welcome any suggestions to further enhance the theoretical contributions of this work.
>
>     From a practical control and coding perspective, we introduce a transformer-based method, Act2Comm, to learn practical policies that enables communication while simultaneously maintaining a desired level of MDP reward. Our experiments demonstrate the effectiveness of Act2Comm in achieving this balance. To further enhance our presentation, we extended the experiments to include a more complex environment and add additional illustrations for the model in the appendix.
>
> * **Weakness 2.** We have corrected the typo and aligned the notations between RL and FSC. The notations such as $s^+$, $s_{t+1}$, $s^{t+1}$ are defined at the end of Section 2, and also noted in the Appendix. The term  $s^+$ is typically used in proofs without explicitly specifying the time step, as it generally represents a successor state or a state in the future. To enhance the readability, we have included a notation table in the Appendix (Table 1), cited in the paper's main body.
>
>   The proof of the main Theorem (Theorem 1) in Appendix 1 relies on the equivalence between the action-state channel and the extended action-state (EAS) channel. Specifically, we demonstrate that the original action-state channel can be equivalently converted to the EAS channel. We initially presented this equivalence in Section 5 for two main reasons: first, the proof of Theorem 1 is provided in the Appendix, and understanding this equivalence is not necessary for comprehending Theorem 1; second, this equivalence is crucial for the design of Act2Comm.
>
>     To address the reviewer's concern and improve readability, we now state this equivalence at the beginning of the proof of Theorem 1 in the Appendix.
>
> * **Weakness 3.**  We believe our study does not fall within the typical domain of information-theoretic RL (IT-RL). To our understanding, IT-RL leverages tools from information theory to enhance the balance between exploration and exploitation. For instance, by focusing on reducing uncertainty and maximizing information gain, IT-RL methods enable agents to learn more effectively in complex and dynamic environments. In contrast, while our work also integrates MDP/RL and information theory, the primary aim is to enable communication between the MDP controller and the receiver, rather than enhancing learning efficiency or robustness.
>
>     We would like to emphasize that mutual information in Theorem 1 is not a statistical measure we impose to regularize our policy, but it appears as the solution of the communication problem whose goal is to maximize the number of bits that can be reliably communicated; that is, it has an operational meaning.
>
>     As the reviewer correctly noted, our problem and the proposed solution are novel, and thus there are no existing benchmarks for direct comparison. The proposed Act2Comm framework is designed for MDPs with finite state and action spaces. Consequently, we chose experimental environments that adhere to these criteria, ensuring that the states are physical signals observable by the receiver (e.g., the position of the pointer in environment 1, the location of the ball and the board in environment 2, the position of the robot in environment 3). There were no additional specific considerations in selecting these experimental environments.

---

> ### Author Response · Authors · 2024-11-21
> **Part 3 (Responses to Questions 1-2)**
>
> * **Question. 1** The general motivation for this study lies in cooperative multi-agent scenarios where communication between agents is beneficial but dedicated communication channels may be unavailable. In such cases, we trade off the rewards of one agent to facilitate communication between this agent and another agent, which can potentially increase the total reward across the entire multi-agent system. For instance, in deep water environments where wireless communication is challenging due to signal attenuation, a team of underwater robots can illustrate this concept. Periodic information exchange is vital for their coordination in task completion, but without dedicated communication links, agents can communicate through actions: the sender encodes messages into its actions, impacting its position, and the receiver decodes the message by observing the sender's position.
>
>      Another motivation may be a scenario where explicit communication is not allowed, or prone to eavesdropping. In such scenarios, communicating through actions can provide a covert way of communication. The encoder may also want to keep its reward as high as possible while secretly communicating, due either to purely pragmatic reasons, or not to be detected. For example, an agent trading in the stock market may not be allowed to communicate with its colluders explicitly, but can convey information secretly through its buy and sell actions. However, it also does not want to sacrifice its gains while communicating.
>
>      We have modified the introduction in the revised manuscript to highlight the potential motivation of the studied scenario. For more examples about the significance of the investigated problem, we refer the reviewer to the first paragraph of the introduction.
>
> * **Question. 2** For any stationary policy $\pi$, let $T_\pi$ and $\rho_\pi$ denote the transition matrix and the stationary state distribution associated with $\pi$, respectively. Then it is well known that $\rho_\pi$ can be calculated by solving the linear equation $\rho_\pi = \rho_\pi T_\pi$, which can be easily solved for any MDPs with finite state and action spaces. The stationary state-action distribution is then given by $\omega_\pi(s,a)=\rho_\pi(s) \pi(a|s)$, where $\pi(a|s)$ denotes the probability of selecting action $a$ in state $s$ by policy $\pi$. In the literature, $\omega_\pi$ is also referred to as the occupation measure of policy $\pi$ (see, e.g., E. Altman, Constrained Markov decision processes. Routledge, 2021). It is also well known that there is a one-to-one correspondence between the set of occupation measures (i.e., the set of state-action distributions) and the set of stationary policies. In particular, each state-action distribution $\omega$ can be translated to a policy $\pi$ via the following formula: \begin{align*}
>          \pi(a|s) = \frac{\omega(s,a)}{\sum_{a'} \omega(s,a')}.
>      \end{align*}
>
>   Therefore, once the stationary state-action distribution is determined by solving the concavity optimization, it can be translated to a stationary policy using the above formula.
>
>      Moreover, as discussed in our response to Weakness 1, the primary significance of Theorem 2 lies in its characterization of the fundamental trade-off between the communication rate through actions and the average reward in MDPs.

---

> ### Author Response · Authors · 2024-11-21
> **Part 4 (Responses to Question 3-4)**
>
> * **Question. 3** To bridge the gap between Sections 4 and 5, we have added additional intuitive descriptions. Meanwhile, the full details and motivation for the model design are provided in the Appendix C.1 for clarity and completeness.
>
>     Intuitions for Section 5:
>
>      (a) Theorem 2 provides an upper bound for practically achievable coding rates under a certain reward constraint. Solving the concave optimization problem yields the capacity and the optimal state-action distribution $\omega$, which can be translated to a stationary policy $\pi$. However, this policy $\pi$ cannot be directly used as the coding policy for communication, as the coding policy needs to generate actions based on both the state and the message. The connection between this and the code proposed in Section 5 is like the relation between Shannon capacity and practical code design in any other communication channel. Capacity provides a theoretical upper bound on the rate of communication in the asymptotic infinite block length regime, but does not tell us how to build practical codes that can approach this bound.
>
>    (b) Our goal in Section 5 is to design a practical channel coding scheme for the action-state channel with feedback in the finite blocklength regime. To effectively map messages and feedback to actions, the transformer architecture with an attention mechanism is naturally well-suited, forming the backbone for both the encoder and decoder components.
>
>     (c) To address the challenge of non-differentiability of the channel, we propose an iterative training strategy that incorporates a critic network to train the encoder and decoder iteratively.
>
>     (d) Instead of directly mapping a message to a sequence of actions, our encoder generates a continuous matrix, which is then quantized into a sequence of decision rules. This design enables block coding, allowing the encoder to determine a sequence of decision rules to be applied over future time steps. Without this approach, actions would need to be generated step-by-step based on the current state, which would significantly increase the computational complexity and reduce the flexibility of the encoding process. Block coding, in contrast, simplifies the process and enhances the encoder’s ability to efficiently plan over extended time horizons.
>
> * **Question. 4** It is correct that, during the training phase, a critic network is trained for $s_{in}$ steps for each update of the encoder. This is necessary due to the non-differentiability of the FSC. While this CriticNet-based approach incurs extra training cost, it is the simplest and most stable method we currently have identified for updating the encoder in our setting. We have also explored other alternative mechanisms with reduced training costs, such as fewer critic training steps, reduced policy noise, and different estimation targets, but they all yielded sub-optimal and unstable results. Details of ablation studies over these alternatives are provided in the Appendix (Table. 5).
>
>     We also want to emphasize that this additional computational cost is incurred only during the offline training process. During the inference phase, the critic network is removed, leaving only the encoder and decoder for policy encoding and information decoding. The detailed illustrations, including figures, pseudocode, and corresponding discussions, are provided in Appendix.C. to enhance the understanding of the paper.
>
>     To better illustrate and accurately quantify the training and inference costs, we provide a complexity analysis of the Act2Comm scheme, detailing the parameters and FLOPs of the model, presented in Appendix C.6, which is also mentioned in the manuscript as:
>
>     ``_Note that this extra training costs is incurred only during the offline training process, this critic network will be removed during the inference phase, as detailed in Appendix C.5-C.6._''

---

> > ### Comment · Reviewer_dvdt · 2024-11-25
> > **Response acknowledgements**
> >
> > I thank the authors for the clear replies. I do not have any other major issue, and I will raise my score accordingly.

---

### Meta-Review · Area_Chair_Kk5w · 2024-12-08

**Metareview:**

This paper introduces an analysis and method enabling reinforcement learning (RL) agents to transmit messages through control actions, represented as sequences of visited states. The approach balances the trade-off between average reward and the information rate of the transmitted message. The authors derive theoretical bounds on the maximum information-theoretic capacity of such transmissions and propose an efficient model and algorithm to achieve this. While there were concerns raised by the reviewers, the rebuttal has helped address the key issues.

**Additional Comments On Reviewer Discussion:**

NA

---

### Decision · Program_Chairs · 2025-01-22

Accept (Poster)